# Predicting Deep Neural Network Generalization with Perturbation Response Curves

**Yair Schiff[1], Brian Quanz[2], Payel Das[2], Pin-Yu Chen[2]**
[1]IBM Watson, [2]IBM Research
{yair.schiff@,blquanz@us.,daspa@us.,pin-yu.chen@}ibm.com

## Abstract

The field of Deep Learning is rich with empirical evidence of human-like performance on a variety of prediction tasks. However, despite these successes, the recent Predicting Generalization in Deep Learning (PGDL) NeurIPS 2020 competition [1] suggests that there is a need for more robust and efficient measures of network generalization. In this work, we propose a new framework for evaluating the generalization capabilities of trained networks. We use perturbation response (PR) curves that capture the accuracy change of a given network as a function of varying levels of training sample perturbation. From these PR curves, we derive novel statistics that capture generalization capability. Specifically, we introduce two new measures for accurately predicting generalization gaps: the Gi-score and Pal-score, which are inspired by the Gini coefficient and Palma ratio (measures of income inequality), that accurately predict generalization gaps. Using our framework applied to *intra* and *inter*-class sample mixup, we attain better predictive scores than the current state-of-the-art measures on a majority of tasks in the PGDL competition. In addition, we show that our framework and the proposed statistics can be used to capture to what extent a trained network is invariant to a given parametric input transformation, such as rotation or translation. Therefore, these generalization gap prediction statistics also provide a useful means for selecting optimal network architectures and hyperparameters that are invariant to a certain perturbation.

## 1 Introduction

Neural networks have produced state-of-the-art and human-like performance across a variety of tasks. This rapid progress has led to wider-spread adoption and deployment. Given their prevalence and increasing applications, it is important to estimate how well a trained net will generalize. Additionally, specific tasks often require models to be invariant to certain transformations or perturbations of the data. This can be achieved either through data augmentation that changes the underlying statistics of the training sets or through inductive architectural biases, such as translation invariance that is inherent in convolutional neural networks. It is important as well to understand how and when a network has been able to learn task-dependent invariances.

Various attempts at bounding and predicting neural network generalization are well summarized and analyzed in the recent survey [2]. While both theoretical and empirical progress has been made, there remains a gap in the literature for an efficient and intuitive measure that can predict generalization given a trained network and its corresponding data *post hoc*. Aiming to fill this gap, the recent Predicting Generalization in Deep Learning (PGDL) NeurIPS 2020 competition [1] encouraged participants to provide *complexity* measures that would take into account network weights and training data to predict generalization gaps, i.e., the difference between training and test set accuracy.

In this work, we propose a new framework that presents progress towards this goal. Our methodology consists of first building an estimate of how the accuracy of a network changes as a function of

35th Conference on Neural Information Processing Systems (NeurIPS 2021).

varying levels of perturbation present in training samples. To do so, we evaluate a trained network's accuracy on a subset of the training dataset that has been perturbed to some degree. Using multiple observations of accuracy vs. perturbation magnitude, we develop a perturbation response (PR) curve for each model. From the PR curves, we derive two new measures called the Gi-score and the Pal-score, which compare a given network's PR curve to that of an idealized network that is unaffected by all perturbation magnitudes. When applying our framework to *inter* and *intra* class Mixup [3] perturbations, we are able to achieve better generalization prediction scores on a majority of the tasks than the current state-of-the-art proposal from the PGDL competition. Because our framework can be applied to any parametric perturbation, we also demonstrate how it can be used to predict the degree to which a network has learned to be invariant to a given perturbation.

## 2 Related work

The PGDL competition resulted in several proposals of complexity measures that aim to predict neural network generalization gaps. While several submissions build off the work of [4] and rely on margin-based measures, we will focus on those submissions that measure perturbation response, specifically to Mixup, since this is most relevant to our work. Mixup, first introduced in [3], is a novel training paradigm in which training occurs not just on the given training data, but also on linearly interpolated points. Manifold Mixup training extends this idea to interpolation of intermediate network representations [5].

Perhaps most closely related to our work is that of the winning submission, [6]. While [6] presents several proposed complexity measures, the authors explore accuracy on Mixup and Manifold Mixup training sets as potential predictors of generalization gap, and performance on mixed up data is one of the inputs into their winning submission. While this closely resembles our work, in that the authors are using performance on a perturbed dataset, namely a mixed up one, the key difference is that [6] only investigates a network's response to a single magnitude of interpolation, 0.5. Additionally, we investigate *between*-class interpolation as well, while [6] only interpolates between data points or intermediate representations within a class, not between classes. Our proposed Gi-score and Pal-score therefore provide a much more robust sense for how invariant a network is to this mixing up perturbation and can easily be applied to other perturbations as well.

In the vein of exploring various transformations / perturbations, the second place submission [7] performs various augmentations, such as color saturation, applying Sobel filters, cropping and resizing, and others, to create a composite penalty score based on how a network performs on these perturbed data points. Our work, in addition to achieving better generalizatiton gap prediction scores, can be thought of as an extension of [7], because as above, rather than looking at a single perturbation level, the Gi-score and Pal-score provide a summary of how a model reacts to a spectrum of parameterized perturbations.

In this work, we also extend our proposed Gi and Pal-scores to predict generalization performance on different data transformations. [8] shows that even for architectures that have inductive biases that should render a network invariant to certain perturbations, in practice these networks are not actually invariant due to how they are trained. This is true when data augmentation is not carefully employed [8], which highlights the need for our predicting invariance line of work. Unlike works, such as [9], that measure how individual layer and neuron outputs respond to input perturbations, e.g., rotation, translation, and color-jittering, we measure how a network's overall accuracy responds to the these perturbations and compare that to an idealized network that is fully invariant. Consistent prediction on invariant data transformations is a desired property for neural networks [10] and it has been widely used as a data augmentation or regularization tool during training for improving generalization [11].

Our scores are inspired by the Gini coefficient and Palma ratio - most commonly used in economics as measures of income inequality [12]–[15]. In economics, the Gini coefficient measures income inequality by ordering a population by income and plotting the percentage of the total national income on the vertical axis vs. the percentage of the population total on the horizontal axis. For an *idealized* economy, wealth distribution lies on a $45°$ line from the origin, which means each percent of the population holds the same percentage of wealth. Plotting this distribution for an actual economy allows for the calculation of the Gini coefficient by taking a ratio of the area between the idealized and actual economy wealth distribution curves and the total area below the idealized economy's curve.

The Palma ratio is also calculated from this plot by taking a ratio of some top $x\%$ of the population's wealth (area below the actual economy's curve) divided by that of some bottom $y\%$.

In addition to income inequality, the Gini coefficient has been used in a wide variety of applications in different domains, e.g, measuring biodiversity in ecology [16], quality of life in health [17], and protein selectivity in chemistry [18]. It is also used in machine learning for measuring classification performance, as twice the area between the ROC curve (curve of false positive rate vs. true positive rate for varying classifier threshold) and the diagonal [19, 20] and has also been used as a criteria for feature selection [21]. Our approach is the first to use this metric together with our PR curve framework for measuring generalization, as well as to use the Pal-score in a machine learning setting.

## 3 Methodology

### 3.1 Notation

We begin by defining a network for a classification task as $f : \mathbb{R}^d \rightarrow \Delta_k$; that is, a mapping of real input signals $x$ of dimension $d$ to discrete distributions, with $\Delta_k$ being the space of all $k$-simplices. We also define the intermediate layer mappings of a network as $f^{(\ell)} : \mathbb{R}^{d_{\ell-1}} \rightarrow \mathbb{R}^{d_\ell}$, where $\ell$ refers to a layer's depth with dimension $d_\ell$. The output of each layer is defined as $x^{(\ell)} = f^{(\ell)}(x^{(\ell-1)})$, with inputs defined as $x^{(0)}$. Additionally, let $f_\ell : \mathbb{R}^{d_\ell} \rightarrow \Delta_k$ be the function that maps intermediate representations $x^{(\ell)}$ to the final output of probability distributions over classes. For a dataset $\mathcal{D}$, consisting of pairs of inputs $x \in \mathbb{R}^d$ and labels $y \in [k]$, a network's accuracy is defined as $\mathcal{A} = \sum_{x,y \in \mathcal{D}} \mathbb{1}(\max_{i \in [k]} f(x)[i] = y) \, / \, |\mathcal{D}|$, i.e. the fraction of samples where the predicted class matches the ground truth label, where $\mathbb{1}(\cdot)$ is an indicator function and $f(x)[i]$ refers to the probability weight of the $i^{\text{th}}$ class.

We define perturbations of the network's representations as $\mathcal{T}_\alpha : \mathbb{R}^{d_\ell} \rightarrow \mathbb{R}^{d_\ell}$, where $\alpha$ controls the magnitude of the perturbation. For example, changing the intensity of an image by $\alpha$ percent can be represented as $\mathcal{T}_\alpha(x^{(0)}) = \alpha x^{(0)}$. To measure a network's response to a perturbation $\mathcal{T}_\alpha$ applied at the $\ell^{\text{th}}$ layer output, we calculate the accuracy of the network for a sample of the training data on which the perturbation has been applied:

$$\mathcal{A}_\alpha^{(\ell)} = \sum_{x,y \sim \mathcal{D}_{sample}} \mathbb{1}(\max_{i \in [k]} f_\ell(\mathcal{T}_\alpha(x^{(\ell)}))[i] = y) \, / \, |\mathcal{D}_{sample}|. \tag{1}$$

The greater the gap $\mathcal{A} - \mathcal{A}_\alpha^{(\ell)}$, the less the network is resilient or invariant to the perturbation $\mathcal{T}_\alpha$ when applied to the $\ell^{\text{th}}$ layer. Perturbations at deeper network layers can be viewed as perturbations in an implicit feature space learned by the network.

### 3.2 Calculating Perturbation Response curves

To measure a network's robustness to a perturbation, one could simply choose a fixed $\alpha$ and measure the network's response. However, a more complete picture is provided by sampling the network's response to various magnitudes of $\alpha$. In Figure 1, we show this in practice. For example, letting $\mathcal{T}_\alpha$ refer to image rotation, we vary $\alpha$ from a minimum $\alpha_{\min}$ degree of rotation, to a maximum $\alpha_{\max}$ degree. For each $\alpha$, we calculate accuracy $\mathcal{A}_\alpha^{(\ell)}$ to measure the network's response to the perturbation of magnitude $\alpha$ applied at depth $\ell$. Plotting $\mathcal{A}_\alpha^{(\ell)}$ on the vertical axis and $\alpha$ on the horizontal axis gives us the PR curves. In Figure 1, we display rotation applied to training images ($\ell = 0$) from the SVHN dataset. This methodology is summarized in Algorithm 1 in Appendix A.3.

### 3.3 Calculating the Gi-score and Pal-score

To extract a single statistic from the PR curves in Figure 1, we draw inspiration from the Gini coefficient and Palma ratio. Namely, we compare a network's response to varying magnitudes of perturbations with an *idealized* network: one whose accuracy is unaffected by the perturbations. The idealized network therefore has a PR curve that starts and remains at accuracy 1.0.

This comparison is achieved by creating a new graph that plots the cumulative density integral under the PR curves against the magnitudes $\alpha_i \in [\alpha_{\min}, \alpha_{\max}]$: $\int_0^{\alpha_i} \mathcal{A}_\alpha d\alpha$. This produces what we call

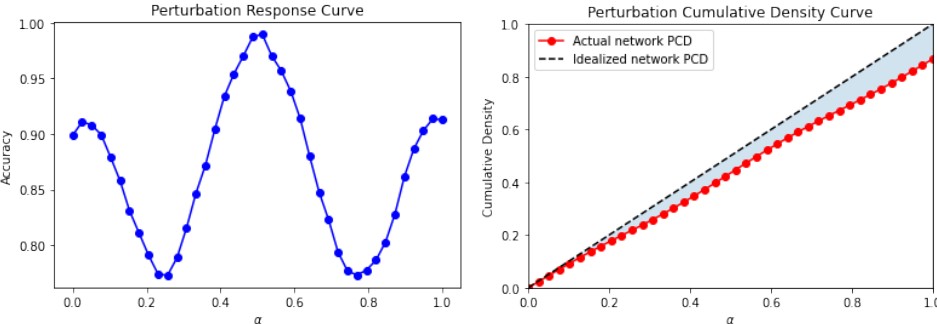

Figure 1: (Left) Sample perturbation response (PR) curve displaying a networks accuracy on 10% of the training data at varying magnitudes of a perturbation. (Right) Sample perturbation cumulative density (PCD) curve comparing the actual network's cumulative density (red line) to that of an idealized network (dotted black line) with shaded area between the curves, which is used in Gi-score calculations. Pal-score comes from the area below the red curve. These sample plots come from a Resnet [22] model trained on SVHN [23] evaluated with rotation as the parameterized perturbation. $\alpha$ varies from $-90°$ to $90°$ of rotation and is normalized in the plots to range from 0 to 1.

*perturbation cumulative density* (PCD) curves seen in Figure 1. For the idealized network whose PR is identically equal to 1 for all $\alpha$, this PCD curve is just the $45°$ line passing through the origin. Finally, the **Gi-score** (named for the Gini coefficient it draws inspiration from) is calculated by taking a ratio of the area between the idealized network's PCD curve and that of the actual network and the total area below the idealized network PCD. We summarize this in Algorithm 2 in Appendix A.4.

The **Pal-score** (named for the Palma ratio it draws inspiration from) calculates the area under the PCD curve and takes a ratio of the area for top 60% of perturbation magnitudes divided by the area for the bottom 10%. This allows us to focus on variations on the upper and lower ends of the perturbation magnitude spectrum, ignoring the middle perturbations that might not vary as widely across networks. We give the pseudocode for the Pal-score in Algorithm 4 in Appendix A.6.

### 3.4 Motivation

Before presenting the empirical effectiveness of our work, we provide detailed intuition underlying our framework. In Economics, the Gini coefficient is specifically designed to compare how the wealth distribution of a given economy compares to that of an idealized economy, where each percentage of the population has an equal share of the nation's income. In that way, the Gini coefficient does not characterize a nation's inequality by simply looking at the percentage of income that an individual percentile of the population holds, rather, the Gini coefficient extracts aggregate information from the entire Lorenz curve which plots the wealth distribution across all percentiles of the population. In our work, this was the main inspiration for creating the Gi-score. We can measure a neural network's response to a certain magnitude of a transformation, but *a priori*, we do not know on which magnitude to focus. We hypothesize that looking at a single value of perturbation magnitude, which is similar to the Mixup approach in [6], may not be ideal and/or informative, and that this can be viewed as a specific case of the more general framework we propose here: evaluating a whole curve of perturbation magnitudes. We therefore propose to examine the spectrum of perturbation-responses and extract a more holistic view of how resilient a network is to a given transformation. A key reasoning behind this is that even if two networks have the same accuracy after the maximum amount of perturbation (e.g., 0.5 for the case of Mixup) that could be a final base level of accuracy deterioration – that multiple networks may even share – but the amount of deterioration before that point could still be different for different networks, and this may also be predictive of generalization behavior (Figure 2).

Justification for using a Pal-score is similar to the reasoning provided above. However, there may be uses cases or domains where there are certain magnitudes of perturbation that are consistent across models, and are therefore uninformative, leading one to focus on certain regions of the PR curves, as with the Palma ratio.

# 4 Experiments

## 4.1 Generalization predictions

To evaluate the extent to which the Gi and Pal-scores accurately predict network generalization, we calculate these statistics on the corpus of trained models provided in the PGDL competition [1]. We use the trained networks and their configurations, training data, and "starting kit" code from the competition; all open-sourced and provided under Apache 2.0 license[1]. The code includes utilities for loading models and model details and running scoring. To this base repository, we added our methods for performing different perturbations at different layers, computing PR curves, and computing our proposed Gi and Pal-scores. Note, we focus on the image modality as this is the type of input data available in the PGDL competition and has been the primary data modality used in prior related work on predicting generalization in other setups as well [2, 4, 24]. Future work will evaluate the application of our framework to other modalities.

### 4.1.1 Generalization predictions: Experimental setup

The networks from this competition contain the following architectures: VGG [25], Network-in-Network (NiN) [26], and Fully Convolutional [27] (Conv) architectures. The datasets are comprised of CIFAR-10 [28], SVHN [23], CINIC-10 [29], Oxford Flowers [30], Oxford Pets [31], and Fashion MNIST [32]. Note, to our knowledge, these datasets are not known to contain personally identifiable information or offensive content. Although CIFAR-10 and CINIC-100 use images from the problematic ImageNet and Tiny Images [33], they contain manually selected subsets. The list of dataset-model combinations, or tasks, available in the trained model corpus can be seen in the first two rows of Table 1. Across the 8 tasks, there are a total of 550 networks. Each network was trained so that it attains nearly perfect accuracy on the training dataset.

As proposed in [1], the goal is to find a *complexity* measure of networks that is causally informative (predictive) of generalization gaps. To measure this predictive quality, [1] propose a Conditional Mutual Information (CMI) score, which measures how informative the complexity measure is about the network's generalization given the network's hyperparameters (i.e., the information contributed by the measure in addition to the network hyperparameters). For full implementation details of this score, please refer to [1] and see Appendix A.2, but roughly, higher values of CMI represent greater capability of a complexity score in predicting generalization gaps.

In our experiments, we let $\mathcal{T}_\alpha$ be defined as an interpolation between two points of either different or the same class: $\mathcal{T}_\alpha(x) = (1-\alpha)x + \alpha x'$, For *inter*-class interpolation, where $x'$ is a (random) input from a different class than $x$, we range $\alpha \in [0, 0.5)$. For the *intra*-class setup, where $x$ and $x'$ are drawn from the same class, we include the upper bound of the magnitude: $\alpha \in [0, 0.5]$. While we explored other varieties of perturbation, such as adding Gaussian noise, we found that this interpolation perturbation was most predictive of generalization gaps for the networks and datasets we tested. Both interpolation perturbations that we test (intra and inter-class) are justifiable for predicting generalization gap. We hypothesize that invariance to interpolation *within* a class should indicate that a network produces similar representations and ultimately the same class maximum prediction for inputs and latent representations that are within the same class regions. Invariance to interpolation *between* classes up to 50% should indicate that the network has well separated clusters for representations of different classes and is robust to perturbations moving points away from heavily represented class regions in the data.

### 4.1.2 Generalization predictions: Results

In Table 1, we present average CMI scores for all models within a task for our Gi and Pal-scores compared to that of the winning team [6] from the PGDL competition. We also compare our statistic to comparable ones presented in [6] that rely on Mixup and Manifold Mixup accuracy[2]. The winning submission described in [6] uses a combination of a score based on the accuracy of mixed up input data and a clustering quality index of class representations, known as the Davies-Bouldin Index (DBI) [34]. Using the notation introduced in Section 3, the measures from [6] present in Table 1

---

[1]https://github.com/google-research/google-research/tree/master/pgdl

[2]Scores from [6] if reported, otherwise we use the author-provided code: https://github.com/parthnatekar/pgdl

can be described as follows: Mixup accuracy: $\mathcal{A}_{0.5}^{(0)}$; Manifold Mixup accuracy: $\mathcal{A}_{0.5}^{(1)}$; DBI * Mixup: $DBI * (1 - \mathcal{A}_{0.5}^{(0)})$.

Table 1: Comparison of Conditional Mutual Information scores for various complexity measures across tasks. We present single-measure scores in the top part of the table and scores based on combinations of multiple measures in the bottom part. For each section, the highest score within a task is bolded. Best scores overall are marked with an asterisk. In CINIC-10 columns, 'bn' stands batch-norm.

| | CIFAR-10 | | SVHN | CINIC-10 | | Oxford Flowers | Oxford Pets | Fashion MNIST | *All Avg* |
|---|---|---|---|---|---|---|---|---|---|
| | *VGG* | *NiN* | *NiN* | *Conv w/bn* | *Conv* | *NiN* | *NiN* | *VGG* | |
| *Single measures only* | | | | | | | | | |
| Gi *inter* $\ell$=0 | 3.03 | **34.34** | 26.58 | 21.01 | 6.96 | 33.05 | **18.46**$^*$ | 4.48 | 18.49 |
| Gi *inter* $\ell$=1 | **7.88** | 22.59 | 12.17 | 12.58 | 8.39 | 7.52 | 4.68 | **16.16**$^*$ | 11.49 |
| Pal *inter* $\ell$=0 | 3.14 | 26.39 | 24.25 | 21.11 | 6.37 | 29.62 | 15.96 | 4.21 | 16.38 |
| Pal *inter* $\ell$=1 | 7.31 | 12.75 | 9.79 | 12.09 | 7.71 | 6.37 | 3.46 | 14.13 | 9.20 |
| Gi *intra* $\ell$=0 | 0.84 | 30.54 | **41.75**$^*$ | 22.97 | 11.46 | **42.44** | 16.21 | 5.10 | **21.41** |
| Gi *intra* $\ell$=1 | 0.22 | 17.18 | 10.96 | 9.50 | 12.43 | 6.92 | 3.60 | 5.55 | 8.29 |
| Pal *intra* $\ell$=0 | 0.61 | 24.36 | 31.82 | 24.15 | 11.01 | 38.10 | 14.04 | 5.12 | 18.65 |
| Pal *intra* $\ell$=1 | 0.44 | 10.34 | 13.48 | 8.68 | 11.09 | 5.88 | 3.02 | 6.25 | 7.40 |
| Mixup | 0.03 | 14.18 | 22.75 | **30.30** | **19.51** | 35.30 | 9.99 | 7.75 | 17.48 |
| Mani. Mix. | 2.24 | 2.88 | 12.11 | 4.23 | 4.84 | 0.03 | 0.13 | 0.19 | 3.33 |
| *Combination measures* | | | | | | | | | |
| PCA Gi&Mi. | 0.04 | 33.16 | 38.08 | **33.76**$^*$ | **20.33**$^*$ | 40.06 | 13.19 | **10.30** | **23.62**$^*$ |
| Pal $\ell$=0*$\ell$=1 | 1.71 | **35.77**$^*$ | **41.58** | 25.14 | 9.50 | 38.92 | **18.41** | 5.61 | 22.08 |
| Pal *inter+intra* | **24.84**$^*$ | 29.70 | 14.04 | 1.64 | 3.45 | 14.84 | 2.13 | 4.89 | 11.94 |
| DBI*Mixup[1] | 0.00 | 25.86 | 32.05 | 31.79 | 15.92 | **43.99**$^*$ | 12.59 | 9.24 | 21.43 |

In the top part of Table 1, we present CMI scores for measures that rely on only one form of Mixup-based measures, comparing Gi and Pal-scores to the one magnitude accuracy predictors from [6]. These results highlight that the Gi-score and Pal-score perform competitively in predicting generalization gap. Note that some versions of our scores out-perform the mixup approaches used in the PGDL winning approach on the majority of tasks, and even substantially out-perform the DBI*Mixup approach on 5/8 tasks. In addition, we believe that our scores provide a more robust measure for how well a model is able to learn invariances to certain transformations. For example, the Mixup complexity score presented in [6] simply takes a 0.5 interpolation of data points and calculates accuracy of a network on the this mixed up portion of the training set. In contrast, our scores allow us to capture network performance on a spectrum of interpolations, thereby providing a more robust statistic for how invariant a network is to linear data interpolation.

Indeed, we found examples of pairs of models that had significantly different generalization gap, but for which Mixup scores were roughly the same, and Gi-score was more reflective of the difference in generalization gap, and show a couple in Figure 2 along with normalized scores and generalization gaps (normalized to be between 0 and 1 across all the models on the data). Looking at the PR curves, we see the higher generalization gap model's PR curve deteriorating more quickly than the lower generalization gap model's PR curve, despite having roughly the same accuracy at $\alpha = 0.5$. As a result, Mixup is not able to capture this difference between the PR curves and model sensitivity to perturbation, unlike our proposed framework and Gi-score which do capture this difference.

We note that for certain architectures different versions of the Gi and Pal-scores seem more useful. Specifically, we observe that for NiN architectures Gi-scores for both inter and intra-class mixup on inputs are most informative, whereas for VGG architectures, Gi and Pal-scores on inter-class mixup from layer 1 representations are better predictors. Future work will explore this pattern to better understand when each score is better suited for predicting generalization. For fully convolutional architectures, both Gi and Pal-scores for intra-class mixup on inputs are good predictors of gener-

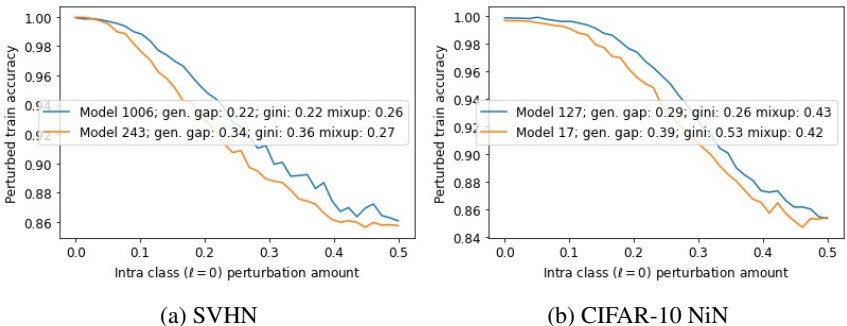

|                |                |
|----------------|----------------|
| (a) SVHN       | (b) CIFAR-10 NiN |

Figure 2: Examples of PR Curves with normalized scores and gen. gaps for 2 different models showing different performance fall-off captured by Gi intra score, but mixup scores roughly the same.

alization gap, however, there is a performance drop compared to the PGDL competition winner on these network types, and Mixup accuracy of one magnitude alone works well on this dataset. It seems for this particular architecture and/or dataset, the deterioration of accuracy at larger perturbations may be more important in predicting generalization. However, even in this case we find that information from other parts of the perturbation response curve can still be useful, as will be described below in our combination scores.

In the bottom part of Table 1, we show CMI scores for a few different ways of combining our Gi and Pal-scores (with different intra vs. inter class interpolation or at different layers) as well as Mixup (intra class interpolation with $\alpha = 0.5$). Additionally we show the PGDL competition winner scores, which itself is a combination method taking the product of Mixup and DBI scores. Scores and details for a larger systematic set of combinations of our scores are provided in Appendix A.8.

In the results shown here, "PCA Gi&Mi." corresponds to performing principal component analysis (PCA) on the pair of Gi intra $\ell=0$ scores and the Mixup scores per task and using the single score derived from projecting the pair of scores on the first principal component. This corresponds to a setting in which we have a set of trained models for a dataset along with the training data and want to estimate their relative generalization performance (as opposed to being applicable to only a single model in isolation). "Pal $\ell=0*\ell=1$" corresponds to taking the product of Pal intra scores at layers 0 and 1, and "Pal *inter+intra*" refers to taking the average of Pal inter and Pal intra scores at layer 0.

We found the best average CMI score (across all tasks) results from combining our Gi intra $\ell=0$ score (that summarizes the whole PR curve) with Mixup (that focuses on the end of the PR curve), via PCA. As mentioned, this seems to suggest the general importance of focusing on multiple parts of the PR curve, paying particular attention to some regions – in this case near the inversion perturbation amount $\alpha = 0.5$ where the perturbed input is potentially farthest away from known examples but still in the same class region, since intra-class interpolation is used.

In general, we found a variety of different combination methods had higher average scores than the competition winner, and also different combinations of scores and methods gave better and sometimes significantly higher scores compared to most other methods for different tasks. For example, simply taking the product of the Pal intra scores at layer 0 and layer 1 gives competitive scores and a higher score on average than the competition winner, and taking the average of Pal inter and Pal intra scores at layer 0 leads to a strikingly higher score on the CIFAR-10 VGG task than most methods (see Appendix A.8 for more examples). This highlights the utility of having our different scores and the potential need to determine what measures work best for a particular architecture type and/or dataset.

### 4.1.3 Generalization predictions: Timing and sensitivity analyses

Our proposed PR curve framework is efficiently computed, as it consists of a simple forward pass of a model with computationally efficient perturbations on a subset of the training data. Here we show results of experiments measuring PR curve generation time and CMI scores as a function of number of random batches of the training data, for different perturbation methods and layers for 2 datasets[3]. For timing analysis on the other tasks, see Appendix A.7. For each batch number experiment, we run

---

[3]PR curve generation takes the most time; score compute time is negligible in comparison (< 1 millisecond)

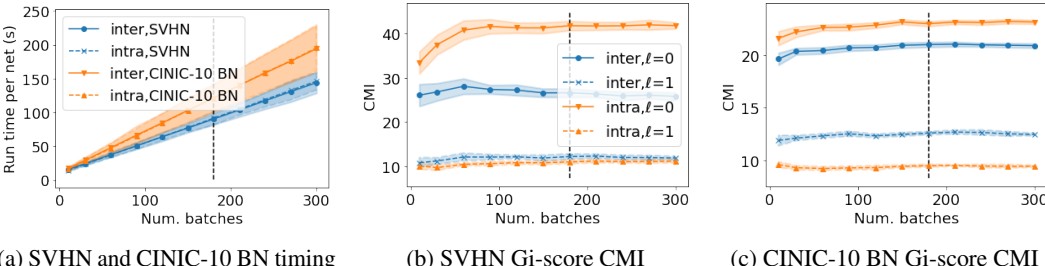

| (a) SVHN and CINIC-10 BN timing | (b) SVHN Gi-score CMI | (c) CINIC-10 BN Gi-score CMI |

Figure 3: Perturbation response curve generation run times (on input layer $\ell = 0$) and Conditional Mutual Information score sensitivity results for 2 datasets. Mean and std. dev. over 20 runs vs. number of batches - 180 batches used in results table (dotted line).

20 random runs and show the mean and standard deviation, to show the sensitivity to the size of the subsample used. Each run is performed with 4 CPUs, 4 GB RAM, and 1 V100 GPU and batch size 128, submitted as resource-restricted jobs to a cluster.

As seen in Figure 3, calculations on the number of batches used in our experiments (180) completes well within the PGDL competition 5 minute time limit (although the competition used 1 K80 GPU per score calculation as opposed to the 1 V100 GPU we used, it also used 4 CPUs as we did and 26GB RAM vs. 4 GB we used). We could have even used more batches and still been within the time limit. This illustrates that the proposed method is a good candidate for common use for evaluating network generalization ability, as it does not add significant computational burden. Additionally the standard deviation of the CMI scores quickly becomes relatively small as the number of batches increases, indicating that these smaller subsamples of the training sets are sufficient to get consistent measure outputs, and that there is small variance in our reported average scores in Table 1.

## 4.2 Measuring invariance

To test whether our framework could accurately predict the degree to which a network has learned a given invariance, we create our own corpus of trained networks.

### 4.2.1 Measuring invariance: Experimental setup

We use two model architectures VGG and ResNet [22] and train both architecture types on the CIFAR-10 and SVHN datasets. For the VGG networks, we use varying depths of 11, 13, 16, and 19 hidden layers with and without batch norm. For the ResNet models, we use varying depths of 18, 34, and 50 hidden layers. For all models, we train with batch sizes of either 1024, 2048, or 4096 and learning rates of either $1e^{-4}$ or $1e^{-5}$. All models are trained with Adam optimization and a learning rate scheduler that reduced learning rate on training loss plateaus.

For each combination of model, dataset, batch norm application, batch size, and learning rate we train and test using four different types of parametric perturbations: (1) rotation, (2) horizontal translation, (3) vertical translation, and (4) color jittering. The $\alpha_{\min}$ and $\alpha_{\max}$ for each perturbation are summarized in Table 2. For rotation, the minimum and maximum perturbation values refer to degree of counter-clockwise rotation. Note, that for SVHN, we use a smaller rotation range than for CIFAR-10, since SVHN contain digits, and we would therefore not want even the *idealized* network to be invariant to the full $360°$ range of rotations. For horizontal translation, the minimum and maximum refer to the amount the image is shifted left or right relative to total image width, with negative values indicating a shift left and positive values indicating a shift right. For vertical translation, the minimum and maximum refer to the amount the image is shifted up or down relative to total image height, with negative values indicating a shift up and positive values indicating a shift down. For color jittering, the minimum and maximum refer to the amount by which brightness, contrast, saturation, and hue are adjusted.

For each perturbation, we train 3 versions of a given hyperparameter combination, one where no data augmentation is used in training, one where partial data augmentation is used in training, and one where full data augmentation is used. Partial data augmentation means that training samples are

Table 2: Perturbation minimum and maximum magnitudes by perturbation type and dataset. Minimum and maximums are displayed in each cell as an ordered pair.

|  | Rotation | Horizontal translation | Vertical translation | Color jittering |
|---|---|---|---|---|
| CIFAR-10 | (-180, 179) | (-0.5, 0.5) | (-0.5, 0.5) | (-0.25, 0.25) |
| SVHN | (-90, 90) | (-0.5, 0.5) | (-0.5, 0.5) | (-0.25, 0.25) |

randomly perturbed with up to 50% of the perturbation minimum and maximum range. Full data augmentation means that training samples are randomly perturbed within the full range.

For all training paradigms, the test set is randomly augmented with the full perturbation range and generalization gap are captured for each model. Naturally, the models that are trained with full data augmentation have better accuracy on the test set, i.e., smaller generalization gap.

Finally, we calculate CMI scores as described in [1] for each architecture type, dataset, and perturbation combination, where the hyperparameters of depth, learning rate, and batch size (and batch norm for VGG networks) are used in the CMI calculation.

### 4.2.2 Measuring invariance: Results

On each dataset, we calculate CMI scores for each of the two architecture types. We repeat this calculation for each of the four perturbation types. For each dataset, model, and perturbation type, we calculate the CMI score for Gi-scores as well as two baselines: (1) the model's accuracy on 10% of the training data that is randomly augmented with the full perturbation range and (2) the mean accuracy for all $\alpha$'s in the PR curve, i.e., $\frac{1}{n_p} \sum_{\alpha_i=\alpha_{\min}}^{\alpha_{\max}} \mathcal{A}_{\alpha_i}^{(0)}$. We chose these baselines as potentially good indicators of invariance to a specific perturbation, in order to see if our statistic provides added predictive capability. The proposed complexity measures from the PGDL competition, such as Mixup from winning team [6], are less relevant in this context as baselines since they do not directly apply to measuring invariance of the perturbations that we test. We only report results for models that were able to achieve at least 80% accuracy on their respective training sets.

Finally, we note that for the perturbation ranges that we test in this section, the PR curves are not monotonically decreasing as they range from a large magnitude negative perturbation to a large magnitude positive perturbation, see for example Figure 1, which shows the PR curve for a rotation perturbation. Therefore, Pal-scores are omitted from this experiment because they are more applicable in cases where the PR curve is roughly monotonically decreasing.

The results of this experiment are presented in Table 3. With higher CMI scores in almost all scenarios, these results highlight that the Gi-score is more informative than simply seeing how the model performs on a randomly augmented subset of the training data. The results also demonstrate that extracting the Gi-score from PR curves is important, as this statistic is also better at predicting generalization gap than mean accuracy of the PR curve, although this naive alternative is also somewhat informative.

## 5 Conclusion

In this work, we introduced a general framework, consisting of computing perturbation response (PR) curves, and two novel statistics, inspired by income inequality metrics, which effectively predict network generalization. The Gi and Pal-scores are intuitive and computationally efficient, requiring only several forward passes through a subset of the training data.

Calculating these statistics on the corpus of data made available in [1] showed that our scores applied to linear interpolation between data points have strong performance in predicting a network's generalization gap. In addition to this predictive capability of generalization, we demonstrated that our framework and the Gi-score provide a useful criterion for evaluating networks and selecting which architecture or hyperparameter configuration is most invariant to a given transformation.

Because they rely on comparison with an idealized network that remains unaffected at all magnitudes of a perturbation, our scores are more informative than other baselines that rely on examining how a trained network responds to perturbed data. Specifically, it is interesting to note that our scores

Table 3: Conditional Mutual Information (CMI) scores for all dataset and model combinations across the four perturbations. Each dataset and model is presented in the columns, and the four perturbations are presented separately. For each perturbation, we report CMI for the Gi-score as well as two baselines: (1) the model's accuracy on 10% of the training data that was randomly augmented with the full perturbation range and (2) the mean accuracy for all $\alpha$'s in the PR curve. Within each perturbation we only consider models that attain at least 80% accuracy on their respective datasets, and we report the sample size of models for each setup along the row that indicates which perturbation we are examining. For each perturbation type, we bold the best CMI within a dataset-model combination. For all but two scenarios, the Gi-score provides the best predictive indicator of generalization gap.

| | CIFAR-10 | | SVHN | |
| | Resnet | VGG | Resnet | VGG |
|---|---|---|---|---|
| *Rotation* | $(n = 34)$ | $(n = 93)$ | $(n = 49)$ | $(n = 142)$ |
| Acc. on augmented train subset | 27.99 | **16.99** | 47.24 | 42.97 |
| Mean acc. on PR curve | 27.61 | 15.61 | 48.14 | 44.05 |
| Gi-score | **41.54** | 15.29 | **54.11** | **46.11** |
| *Horizontal translation* | $(n = 36)$ | $(n = 112)$ | $(n = 50)$ | $(n = 143)$ |
| Acc. on augmented train subset | 41.79 | 33.48 | 29.49 | 24.20 |
| Mean acc. on PR curve | 45.03 | 33.00 | 29.88 | 24.28 |
| Gi-score | **50.07** | **34.31** | **34.56** | **25.94** |
| *Vertical translation* | $(n = 36)$ | $(n = 107)$ | $(n = 49)$ | $(n = 141)$ |
| Acc. on augmented train subset | 26.79 | 35.68 | 51.88 | 50.98 |
| Mean acc. on PR curve | 26.55 | 37.33 | 52.39 | 52.22 |
| Gi-score | **34.85** | **39.07** | **59.02** | **52.83** |
| *Color-jittering* | $(n = 44)$ | $(n = 130)$ | $(n = 49)$ | $(n = 143)$ |
| Acc. on augmented train subset | 35.77 | 28.44 | 43.08 | **30.07** |
| Mean acc. on PR curve | 39.12 | 29.37 | 43.26 | 28.67 |
| Gi-score | **44.63** | **30.79** | **50.08** | 29.45 |

provide more informative measures than simply averaging the PR curves or taking the accuracy for random perturbation magnitudes – suggesting our scores may better capture varying effects of different perturbation amounts (perhaps related to discoveries presented in [35] for augmenting training). Future work will explore this hypothesis further. Future work will also demonstrate the usefulness of our statistics on other parametric perturbations and will compare the Gi and Pal-scores of architectures that are known to be more invariant to certain perturbations, e.g., those described in [10], to those for models without the inductive biases that contribute to invariance.

Although we provide strong empirical evidence that our framework and Gi and Pal-scores are distinctly informative of network generalization and invariance, currently our approach lacks theoretical underpinning of the performance obtained by these scores. Future work includes deriving theoretical analyses detailing under which conditions our scores correctly identify the best network and correctly rank networks based on generalization, as well as consistency guarantees. Additionally, we plan to evaluate our framework on other tasks and modalities, such as language models and time series. We also plan to explore the possibility of regularization approaches based on our scores.

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
