# A Appendix

## A.1 Assets

For the generalization prediction experiments, we use the data provided by the PGDL competition organizers [1], which is available for download here: https://github.com/google-research/google-research/tree/master/pgdl#wheres-the-data and is released under the Apache 2.0 license, and corresponding Tensorflow [36] code for loading models, and wrote our perturbation code using Tensorflow as well.

For the measuring invariance experiments, we use the CIFAR-10 [28] dataset, released under the MIT license, and SVHN dataset [23], which does not have a license listed with the dataset. Both datasets were downloaded and split into training and test sets by the torchvision module of the PyTorch [37] library.

The packages used in this work and their respective licenses are listed below:

1. PGDL Competition Starter Kit [1]; Apache 2.0
2. PyTorch [37]; BSD
3. PyTorch Lightning [38]; Apache 2.0
4. Tensorflow [36]; Apache 2.0

## A.2 Calculating Conditional Mutual Information scores

Given it's importance to our analyses, we reproduce the calculation for Conditional Mutual Information (CMI) scores presented in [1] here.

Generalization gap is defined as:

$$g(f, \mathcal{D}) = \sum_{x,y \in \mathcal{D}_{train}} \mathbb{1}(\max_{i \in [k]} f(x)[i] = y) \, / \, |\mathcal{D}_{train}| - \sum_{x,y \in \mathcal{D}_{test}} \mathbb{1}(\max_{i \in [k]} f(x)[i] = y) \, / \, |\mathcal{D}_{test}|$$

The goal of the PGDL competition was to find a complexity measure $\mu$ such that:

$$\text{sgn}(\mu(f, \mathcal{D}_{train}), \mu(f', \mathcal{D}_{train})) = \text{sgn}(g(f, \mathcal{D}), g(f', \mathcal{D}))$$

Let

$$V_g(f, f') = \text{sgn}(g(f, \mathcal{D}), g(f', \mathcal{D}))$$

and

$$V_\mu(f, f') = \text{sgn}(\mu(f, \mathcal{D}_{train}), \mu(f', \mathcal{D}_{train}))$$

Now, we use $\mathcal{O}$ to denote the set of hyperparameters. For example, in Section 4.2, for Resnet models we use $\mathcal{O} = \{\text{depth}, \text{learning rate}, \text{batch size}\}$ so that $|\mathcal{O}| = 3$. Models can be separated into groups based on their specific value for each hyperparameter. Each group is denoted as $\mathcal{O}_k$, i.e., the set of models that have hyperparameter configuration $k$. For each hyperparameter, $\Theta_i \in \mathcal{O}$, $|\Theta_i|$ is the number of possible values that parameter can take on, so that for example, $\Theta_i = $ Resnet depths of 18, 34, 50, we have $|\Theta_i| = 3$. Thus the total number of groups is $\prod_{\Theta_i \in \mathcal{O}} |\Theta_i|$.

If we treat $V_g$ and $V_\mu$ as Bernoulli random variables, then we can calculate the probabilities:

$$p(V_g | \mathcal{O}_k), \ \ p(V_\mu | \mathcal{O}_k), \ \ p(V_g, V_\mu | \mathcal{O}_k)$$

where the probabilities are calculated by counting over models in each group $\mathcal{O}_k$.

With these probabilities, we can define mutual information between $V_g$ and $V_\mu$:

$$\mathcal{I}(V_g, V_\mu | \mathcal{O}_k) = \sum_{V_g} \sum_{V_\mu} p(V_g, V_\mu | \mathcal{O}_k) \log \left( \frac{p(V_\mu, V_g | \mathcal{O})}{p(V_g | \mathcal{O}_k) p(V_\mu | \mathcal{O}_k)} \right)$$

Each $\mathcal{O}_k$ occurs with the same probability $p_c = 1 / \prod_{\Theta_i \in \mathcal{O}} |\Theta_i|$. Thus, using the same notation abuse as in [1], we have mutual information between $V_\mu$ and $V_g$ conditioned on the values of $\mathcal{O}$:

$$\mathcal{I}(V_g, V_\mu | \mathcal{O}) = \sum_{\mathcal{O}_k} p_c \mathcal{I}(V_g, V_\mu | \mathcal{O}_k)$$

To get values between 0 and 1, we can normalize this conditional mutual information by conditional entropy of generalization, which is defined as:

$$\mathcal{H}(V_g|\mathcal{O}) = \sum_{\mathcal{O}_k} p_c \sum_{V_g} \log(p(V_g|\mathcal{O}_k))$$

So we now have normalized conditional mutual information:

$$\hat{\mathcal{I}}(V_g, V_\mu|\mathcal{O}) = \frac{\mathcal{I}(V_g, V_\mu|\mathcal{O})}{\mathcal{H}(V_g|\mathcal{O})}$$

Finally, the CMI score used in PGDL and presented in the results sections in this work is defined as:

$$\text{CMI}(\mu) = \min_{\mathcal{O}} \hat{\mathcal{I}}(V_g, V_\mu|\mathcal{O})$$

### A.3 Algorithm for generating Perturbation Response Curves

Here we provide the detailed algorithm pseudo-code for generating perturbation response curves, described in the main paper.

---

**Algorithm 1:** Building Perturbation Response (PR) Curve

---

**Inputs:** Trained model $f$; Dataset $\mathcal{D}$; Perturbation $\mathcal{T}_\alpha$; Min perturbation magnitude $\alpha_{\min}$; Max perturbation magnitude $\alpha_{\max}$; Number of perturbation magnitudes to measure $n_p$; Layer at which to apply the perturbation $\ell$; number of batches to sample $n_b$; batch size $b_s$

**Output:** PR Curve: Arrays of regularly spaced perturbation magnitudes ranging from $\alpha_{\min}$ to $\alpha_{\max}$ of length $n_p$ $[\alpha_{\min}, \alpha_{\max}][n_p]$ and accuracy array at each perturbation magnitude of length $n_p$ $\mathcal{A}_\alpha[n_p]$

**for** $i \leftarrow 0$ *to* $n_p - 1$ **do**
    $\alpha_i \leftarrow [\alpha_{\min}, \alpha_{\max}][i]$
    Shuffle $\mathcal{D}$
    **for** $k \leftarrow 0$ *to* $n_b - 1$ **do**
        $\mathcal{D}_{sample} \leftarrow \mathcal{D}[kb_s : (k+1)b_s]$ // batch $k$ of $\mathcal{D}$
        $\mathcal{A}_{\alpha_i}^{(\ell)}[\text{k}] \leftarrow$ batch accuracy under perturbation $\mathcal{T}_{\alpha_i}$ (Equation 1)
    $\mathcal{A}_\alpha[\text{i}] \leftarrow \sum_k \mathcal{A}_{\alpha_i}^{(\ell)}[\text{k}]/n_b$

---

### A.4 Algorithm for Gi-scores

Here we provide the detailed algorithm described in the main paper for computing the Gi-scores, shown in Algorithm 2.

### A.5 Algorithm for efficiently computing interpolation PR curves

Here we give more details about how we compute the accuracy under interpolation per batch – for the intra class interpolation. In order to do this efficiently, we compute accuracy per randomly sampled batches with simple operations that can be encoded in a computational graph (e.g., Tensorflow), and then compute the mean accuracy from multiple batch accuracies. Inter-class interpolation is performed similarly. In each case, we split the batch into pairs efficiently, by random pairing for inter-class interpolation since probability of being from different classes is high, and by sorting by class label and then interleaving the sorted entries to get the pairs, since it is most likely to get pairs of the same class this way. In both cases we throw out any pairs from the batch that do not match (any pairs having the same class for inter-class interpolation and any pairs having different class for intra-class interpolation). We then compute the accuracy for the batch, and keep track of the effective batch size, to update the total accuracy across all batches.

### A.6 Algorithm for computing Pal-scores

In this section, we present the pseudocode for calculating the Pal-score, in Algorithm 4, which is similar to the Gi-score calculation, but with a focus on certain areas of the PCD curve.

---

**Algorithm 2:** Gi-Score computation given PR Curve for a model

---

**Inputs:** Arrays of perturbation magnitude $\alpha[n]$ and accuracy $\mathcal{A}_\alpha[n]$
**Output:** Gi-score $gi$
$a_t[0] \leftarrow 0$ `// initialize 1st element of trapezoidal areas array with 0`
**for** $i \leftarrow 0$ *to* $n - 2$ **do**
    $a_t[i + 1] \leftarrow 0.5(\alpha[i + 1] - \alpha[i])(\mathcal{A}_\alpha[i] + \mathcal{A}_\alpha[i + 1])$
**for** $i \leftarrow 1$ *to* $n - 1$ **do**
    $a_t[i] \leftarrow a_t[i] + a_t[i - 1]$. `// cumulative sum`
$d[i] = \alpha[i] - a_t[i], \forall i$
$gi = 0$
**for** $i \leftarrow 0$ *to* $n - 2$ **do**
    $gi \leftarrow gi + 0.5(\alpha[i + 1] - \alpha[i])(d[i] + d[i + 1])$
$gi \leftarrow gi/(0.5\alpha[n - 1]^2)$ `// Divide by area under line of equality`
**return** $gi$

---

---

**Algorithm 3:** Efficient computation of intra-class interpolation for a batch

---

**Inputs:** Sampled batch of data for layer $\ell$ to perturb $x^{(\ell)}, y$ of size $n$ (assume here $n$ is an even
         number to simplify description), perturbation magnitude $\alpha \in [0, 1]$, and network
         function $f_{\ell+1}$
**Output:** Accuracy for batch sample $\mathcal{A}_\alpha^{(\ell)}$
Sort $x^{(\ell)}, y$ by class labels $y$
`// Assign every other element, starting with first element`
$x_1^{(\ell)} \leftarrow x^{(\ell)}[:: 2]$
$y_1 \leftarrow y[:: 2]$
`// Assign every other element, starting with second element`
$x_2^{(\ell)} \leftarrow x^{(\ell)}[1 :: 2]$
$y_2 \leftarrow y[1 :: 2]$
`// Each entry at index` $i$ `in` $x_1^{(\ell)}$ `and` $x_2^{(\ell)}$`, and` $y_1$ `and` $y_2$ `form pairs`
Drop any index $i$ in $x_1^{(\ell)}, x_2^{(\ell)}, y_1$, and $y_2$ where $y_1[i] \neq y_2[i]$
$x_p^{(\ell)} \leftarrow (1 - \alpha)x_1^{(\ell)} + \alpha x_2^{(\ell)}$ `// Get interpolated point`

`// Compute mean accuracy for interpolated points`
$\mathcal{A}_\alpha^{(\ell)} = \sum_j \mathbb{1}(\max_{i \in [k]} f_{\ell+1}(x_p^{(\ell)}[j])[i] = y_1[j]) / |y_1|$
**return** $\mathcal{A}_\alpha^{(\ell)}$

---

## A.7 Additional sensitivity analysis

Here we show the timing (Figure 4) and Gi-score CMI (Figure 5) sensitivity results for all datasets for both intra and inter-class perturbations, and for both input ($\ell = 0$) and first hidden layer ($\ell = 1$). This analysis is averaged over 20 different runs (each having different random sampling of the full data set per batch) – with mean and standard deviation shown.

The first set of plots shows the timing results for all datasets (Figure 4). We see that for plots, even those with larger number of batches, the time taken to compute our scores is well under the competition time limit – and the longest time dataset (CINIC-10) was already reported in the main paper.

Figure 5 shows the Gi-score CMI sensitivity to number of batches (mean and std. dev.) for all datasets, for both perturb types (inter and intra-class) and for both layers (input - layer 0, and layer 1). This confirms the stability of the scores with sufficient sub-sample size (number of batches) and that the number of batches was chosen to be large enough to ensure stable scores in our reported results.

**Algorithm 4:** Pal-Score computation given PR Curve for a model

**Inputs:** Arrays of perturbation magnitude $\alpha[n]$ and accuracy $\mathcal{A}_\alpha[n]$
**Output:** Pal-score $pal$
$a_t[0] \leftarrow 0$ // initialize 1st element of trapezoidal areas array with 0
**for** $i \leftarrow 0$ *to* $n - 2$ **do**
$\quad \lfloor \quad a_t[i + 1] \leftarrow 0.5(\alpha[i + 1] - \alpha[i])(\mathcal{A}_\alpha[i] + \mathcal{A}_\alpha[i + 1])$
$top\_idx \leftarrow$ index of 60% of $\alpha[n]$
$bottom\_idx \leftarrow$ index of 10% of $\alpha[n]$
$pal \leftarrow a_t[top\_idx]/a_t[bottom\_idx]$
**return** $pal$

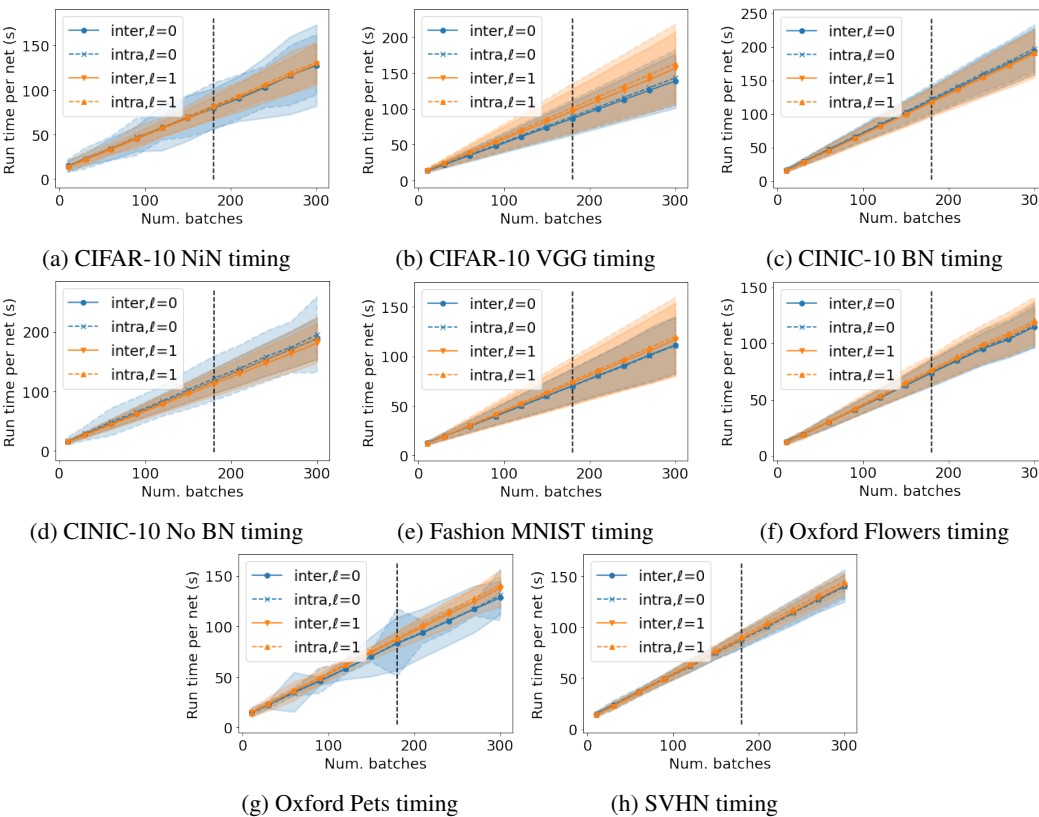

(a) CIFAR-10 NiN timing     (b) CIFAR-10 VGG timing     (c) CINIC-10 BN timing

(d) CINIC-10 No BN timing     (e) Fashion MNIST timing     (f) Oxford Flowers timing

(g) Oxford Pets timing     (h) SVHN timing

Figure 4: Perturbation response curve generation run times for each datasets on input layer and layer 1 ($\ell = 0$ and $\ell = 1$). Mean and std. dev. over 20 runs vs. number of batches – 180 batches used in results table (dotted line)

## A.8 Measuring generalization: Complete GI and Pal Score combination results

In this section, we show results for a complete systematic set of combinations of our different measures (Gi and Pal-scores with different intra and inter-class interpolation, and at different layers, 0 and 1) as well as Mixup (intra-class interpolation with $\alpha = 0.5$) in select cases, for different score combination approaches. This is not an exhaustive set of combinations, but fairly broad coverage of possible combinations, using only relatively simple approaches.

We also include two types of combinations that require a set of different models / complete set of models for a task before the final score can be derived to estimate generalization: principle component analysis (PCA) and rank-based combinations. This could correspond to the setting in which we have a set of trained models for a dataset along with the training data and want to estimate their relative generalization performance (as opposed to being applicable to only a single model). Details of how

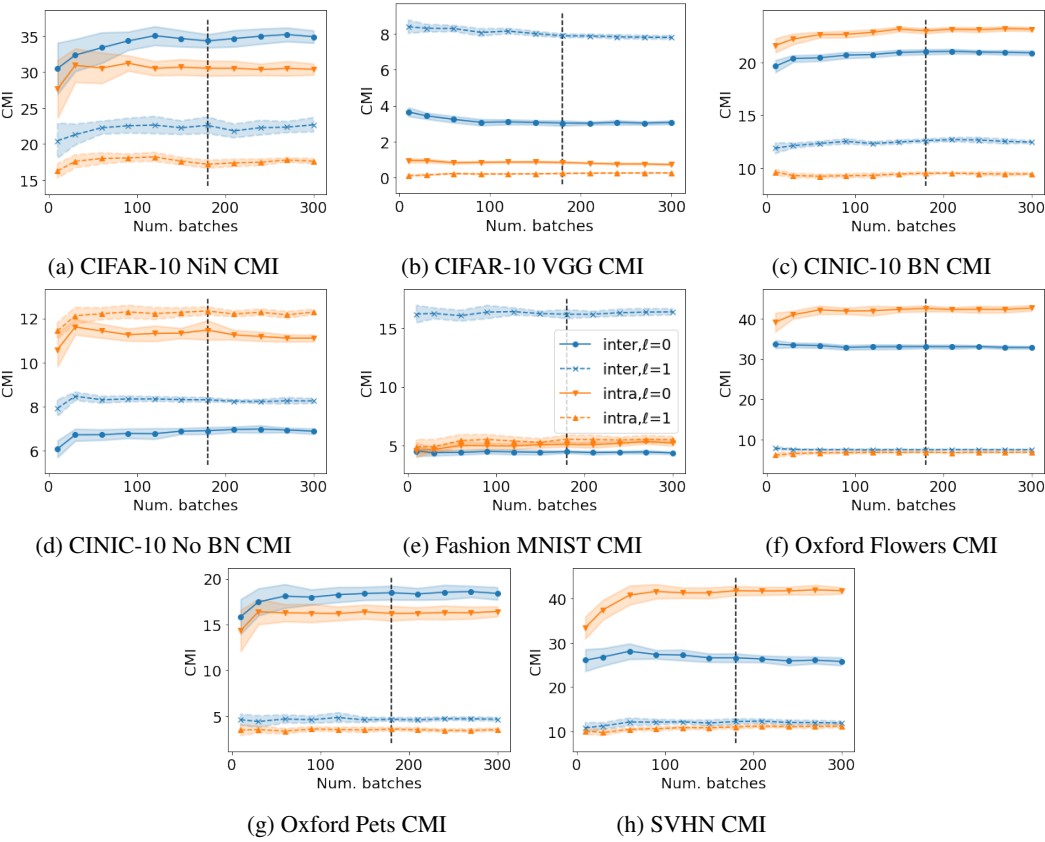

Figure 5: Conditional Mutual Information (CMI) score sensitivity results for all datasets on input layer and layer 1 ($\ell = 0$ and $\ell = 1$). Mean and std. dev. over 20 runs vs. number of batches – 180 batches used in results table (dotted line)

the combined scores are computed with these two approaches are given in the following sections along with the other combination methods that do not have this constraint.

The types of combinations we used are the following, which are described in detail in the subsequent sections, along with corresponding results tables:

- PCA and NPCA

- AVG

- PROD

- PROD+AVG

- AVG RANK

Note, for any combination of Gi and Pal-scores, we use the negative of the Pal-scores, since Gi and Pal-scores are oppositely correlated (anti-correlated). Therefore, in order to support all forms of combining the scores, we use the negative of Pal-score when combining with Gi-score, as some combination methods would not otherwise work as expected if they are oppositely correlated, e.g., taking the average of the two scores.

It is also interesting to note that for any type of combination, there are some combinations that work well on average across tasks – i.e., better than the competition winner average score across tasks, but different combinations of different methods work best on different tasks.

### A.8.1 PCA and NPCA (Tables 4 and 5 resp.)

Given the full set of models and scores for each task, to get the combined score we perform principal component analysis (PCA) on the set of scores being combined and project the multiple scores on the first principal component to get a single combination score for each model. The intuition is to combine the scores by capturing the principal direction of variation amongst the two or more score dimensions, for each task. We combine pairs of various Gi and Pal=scores, as well as Gi-score with Mixup, and also include combinations of all Gi and Pal-scores ("PCA all Gi & Pal"), all Gi-scores ("PCA all Gi") and all Pal-scores ("PCA all Pal"). We also include in the second table the results using PCA after normalizing / standardizing the individual scores ("NPCA") across the models in each task, i.e., subtracting the mean and dividing by standard deviation.

As shown in the main paper, the combination that gives the best average score comes from combining Gi intra at level 0 with Mixup. Various combinations do better at different tasks. The fact that combining the Gi-score and Mixup at level 0, both with intra-class interpolation, does best on average may suggest that it is important to pay attention to both how quickly the PR curve falls off for varying amounts of perturbation as well as how much it falls off near the inversion point of $\alpha = 0.5$, where it is farthest away from any training example. This may encourage developing alternate scores from the PR curves as well.

It is also interesting to note that combining more than 2 methods in this way, such as all Gi and Pal-scores, does not give the best results or better scores in general than combining pairs. This may be because with more scores included, the most degree of variation can come from differences in the scores that are not connected to generalization, and choosing only one principal component dimension limits what is captured from the multiple scores. Without supervisory information (i.e., test errors) the PCA components found may not line up with generalization in general. In general, combining multiple scores in an unsupervised way can be a challenging task, so another possibility for future work is looking for ways to combine scores given some labeled training data (in the form of models with generalization gaps known for given datasets) such that these combinations also work for unseen tasks (datasets and models). This echoes the discussion above, as learning how to combine the scores from different areas of the PR curve may give the best results, as might be suggested by seeing good results with the combination of Gi intra $\ell 0$ and Mixup.

### A.8.2 AVG (Table 6)

For these set of results, we simply average the two scores together. In this case, a simple average of the Gi intra score at level 0 and the Pal intra score at level 0 (again taking the negative of the latter since it is anti-correlated with Gi-score) yields the best average score across tasks.

It is also interesting to note that certain combinations of the scores across layers and interpolation types give significantly better scores compared to most other methods for certain specific tasks. For instance, combining Gi intra $\ell 0$ and Gi inter $\ell 1$ gives a higher score on Fashion MNIST, and combining Pal intra $\ell 0$ and Pal inter $\ell 0$ gives a strikingly higher score on the CIFAR-10 VGG task compared to most other methods.

### A.8.3 PROD (Table 7)

For this combination approach, we take the product of the two scores being combined. In this case the product of Pal intra $\ell 0$ and Pal inter $\ell 0$ gave the best results on average.

### A.8.4 PROD+AVG (Table 8)

This combination approach combines the previous two. The product of the two scores being combined is added to their average, to obtain the final single score.

### A.8.5 AVG RANK (Table 9)

For this and the next combination method, raw scores for each individual score are first transformed to ranks per task (model and dataset combination), by ranking the scores from smallest to largest, and assigning the smallest rank a score of 1 and the largest a score of the number of models in the task. For AVG RANK, the average is taken between the rank-transformed scores of two different scores.

Again, if a Gi-score is combined with a Pal-score, the negative of the Pal-score is used, since Gi and Pal-scores are anti-correlated.

For this AVG RANK approach, Gi intra $\ell0$ combined with Pal intra $\ell0$ gave the best average result, suggesting some useful and different information is captured by the different score approaches, so combining them can be useful.

Table 4: Comparison of Conditional Mutual Information scores for various complexity measures across tasks. We present combinations of multiple of our measures using PCA per task, and include the PGDL competition winner scores at the bottom. The highest score within each task is bolded. In the CINIC-10 columns, 'bn' stands batch-norm.

| | CIFAR-10 | | SVHN | CINIC-10 | | Oxford Flowers | Oxford Pets | Fashion MNIST | *All Avg* |
|---|---|---|---|---|---|---|---|---|---|
| | *VGG* | *NiN* | *NiN* | *Conv w/bn* | *Conv* | *NiN* | *NiN* | *VGG* | |
| PCA all Gi & Pal | 5.08 | 28.24 | 18.22 | 19.80 | 10.71 | 20.10 | 5.03 | 7.36 | 14.32 |
| PCA all Gi | 4.98 | 25.49 | 17.89 | 17.42 | 9.84 | 20.30 | 6.56 | 6.25 | 13.59 |
| PCA all Pal | 5.08 | 28.24 | 18.22 | 19.80 | 10.71 | 20.10 | 5.03 | 7.36 | 14.32 |
| PCA Gi intra $\ell0$ & Gi intra $\ell1$ | 0.11 | 25.51 | 19.06 | 16.18 | 12.14 | 16.02 | 6.37 | 5.38 | 12.60 |
| PCA Gi intra $\ell0$ & Gi inter $\ell0$ | 2.02 | 33.33 | 35.06 | 22.24 | 8.95 | 37.70 | 17.05 | 4.55 | 20.11 |
| PCA Gi intra $\ell0$ & Gi inter $\ell1$ | 6.24 | 24.91 | 21.83 | 17.84 | 9.64 | 12.32 | 6.20 | 16.68 | 14.46 |
| PCA Gi intra $\ell0$ & Pal intra $\ell0$ | 0.88 | 35.85 | **43.51** | 26.72 | 12.89 | **43.44** | 18.00 | 7.02 | 23.54 |
| PCA Gi intra $\ell0$ & Pal intra $\ell1$ | 0.25 | 15.62 | 12.35 | 9.89 | 13.07 | 7.02 | 5.23 | 8.32 | 8.97 |
| PCA Gi intra $\ell0$ & Pal inter $\ell0$ | 3.33 | **37.22** | 32.37 | 23.90 | 7.69 | 29.62 | 17.14 | 5.35 | 19.58 |
| PCA Gi intra $\ell0$ & Mix intra $\ell0$ | 0.04 | 33.16 | 38.08 | **33.76** | **20.33** | 40.06 | 13.19 | 10.30 | **23.62** |
| PCA Gi intra $\ell1$ & Gi inter $\ell0$ | 7.77 | 26.67 | 16.41 | 16.31 | 9.76 | 28.29 | 6.30 | 4.52 | 14.50 |
| PCA Gi intra $\ell1$ & Mix intra $\ell0$ | 0.01 | 32.38 | 32.71 | 32.97 | 19.92 | 39.59 | 13.96 | 10.78 | 22.79 |
| PCA Gi inter $\ell0$ & Gi inter $\ell1$ | **8.05** | 25.93 | 19.92 | 16.92 | 8.30 | 20.41 | 6.46 | 5.86 | 13.98 |
| PCA Gi inter $\ell0$ & Mix intra $\ell0$ | 0.49 | 33.52 | 38.61 | 33.72 | 18.94 | 40.96 | 13.63 | 5.36 | 23.16 |
| PCA Gi inter $\ell1$ & Mix intra $\ell0$ | 0.10 | 30.08 | 34.92 | 33.46 | 18.50 | 36.20 | 13.12 | 16.95 | 22.91 |
| PCA Pal intra $\ell0$ & Pal inter $\ell0$ | 2.36 | 36.18 | 42.26 | 25.14 | 9.82 | 38.43 | **18.35** | 5.50 | 22.26 |
| PCA Pal intra $\ell0$ & Pal inter $\ell1$ | 6.79 | 23.88 | 21.05 | 21.04 | 10.60 | 13.82 | 4.87 | **17.12** | 14.90 |
| *DBI*Mixup* | *0.00* | *25.86* | *32.05* | *31.79* | *15.92* | *43.99* | *12.59* | *9.24* | *21.43* |

## A.9 Measuring invariance: Additional experimental setup information

In this section, we provide additional details about the measuring invariance experiments.

For training the Resnet and VGG networks on CIFAR-10 and SVHN, we rely on the the the PyTorch framework [39] and the PyTorch-Lightning wrapper [38]. In order to mimic a real world training paradigm, we split the training sets for both CIFAR-10 and SVHN into 95% training data and 5% validation data.

Each dataset, model, perturbation experiment combination is performed with 1 CPU and 1 V100 GPU.

## A.10 Measuring invariance: Additional results

In this section, we include scatter plots of model generalization gap vs. our statistics and the baselines from the measuring invariance experiments.

In each plot, we also include the number of models $n$, the CMI score, and the Pearson R correlation coefficient. Results are displayed in Figures 6, 7, 8, and 9. Note that, interestingly, correlation and visual inspection do not always completely equate with CMI scores, as CMI specifically measures information provided by the complexity measure beyond what is known given the network hyperparameters / settings. In particular, when the factors conditioned on correlate with generalization gap, the differentiating contribution of the complexity measure itself may not be as easily observable, and the CMI score is needed to fully elucidate the complexity measures' distinct predictive / informative capabilities. We observe that our approach is best able to distinguish the different training condition

Table 5: Comparison of Conditional Mutual Information scores for various complexity measures across tasks. We present combinations of multiple of our measures using PCA per task after normalizing / standardizing each score per task (subtracting mean and dividing by std. dev. across the task), and include the PGDL competition winner scores at the bottom. The highest score within each task is bolded. In the CINIC-10 columns, 'bn' stands batch-norm.

| | CIFAR-10 | | SVHN | CINIC-10 | | Oxford Flowers | Oxford Pets | Fashion MNIST | All Avg |
|---|---|---|---|---|---|---|---|---|---|
| | *VGG* | *NiN* | *NiN* | *Conv w/bn* | *Conv* | *NiN* | *NiN* | *VGG* | |
| NPCA Gi intra ℓ0 & Gi intra ℓ1 | 0.31 | 27.49 | 17.92 | 16.13 | 12.12 | 15.65 | 10.08 | 4.83 | 13.07 |
| NPCA Gi intra ℓ0 & Gi inter ℓ0 | 1.20 | 32.38 | 34.27 | 22.33 | 8.79 | 40.31 | 17.03 | 5.16 | 20.18 |
| NPCA Gi intra ℓ0 & Gi inter ℓ1 | 2.68 | 29.00 | 19.63 | 18.41 | 9.64 | 23.20 | 10.71 | 11.93 | 15.65 |
| NPCA Gi intra ℓ0 & Pal intra ℓ0 | 0.85 | 31.94 | **43.06** | 25.01 | 12.15 | **43.52** | 17.07 | 6.10 | 22.46 |
| NPCA Gi intra ℓ0 & Pal intra ℓ1 | 0.34 | 28.03 | 17.68 | 16.54 | 12.63 | 16.07 | 10.99 | 6.44 | 13.59 |
| NPCA Gi intra ℓ0 & Pal inter ℓ0 | 1.24 | 33.23 | 38.39 | 23.53 | 8.74 | 40.56 | 17.70 | 5.35 | 21.09 |
| NPCA Gi intra ℓ0 & Mix intra ℓ0 | 0.21 | 33.05 | 41.86 | **30.29** | **18.87** | 42.47 | 16.08 | 7.78 | **23.83** |
| NPCA Gi intra ℓ1 & Gi inter ℓ0 | 6.07 | 26.82 | 16.44 | 15.95 | 9.90 | 14.93 | 10.05 | 6.12 | 13.29 |
| NPCA Gi intra ℓ1 & Mix intra ℓ0 | 0.07 | 29.17 | 16.78 | 22.69 | 18.45 | 15.39 | 10.40 | 8.22 | 15.15 |
| NPCA Gi inter ℓ0 & Gi inter ℓ1 | **7.62** | 28.69 | 18.94 | 17.05 | 8.23 | 21.20 | 10.47 | 8.61 | 15.10 |
| NPCA Gi inter ℓ0 & Mix intra ℓ0 | 1.79 | 35.07 | 39.08 | 30.05 | 15.40 | 40.30 | 16.87 | 8.03 | 23.33 |
| NPCA Gi inter ℓ1 & Mix intra ℓ0 | 1.78 | 29.93 | 18.34 | 26.84 | 15.48 | 23.82 | 10.24 | **16.30** | 17.84 |
| NPCA Pal intra ℓ0 & Pal inter ℓ0 | 1.38 | **35.39** | 41.46 | 25.20 | 9.51 | 40.44 | **18.18** | 7.28 | 22.36 |
| NPCA Pal intra ℓ0 & Pal inter ℓ1 | 2.58 | 31.28 | 19.19 | 21.37 | 10.60 | 24.27 | 9.23 | 14.12 | 16.58 |
| *DBI*Mixup* | *0.00* | *25.86* | *32.05* | *31.79* | *15.92* | *43.99* | *12.59* | *9.24* | *21.43* |

Table 6: Comparison of Conditional Mutual Information scores for various complexity measures across tasks. We present combinations of our measures using the simple average of two scores, and include the PGDL competition winner scores at the bottom. Note in this case, since Gi and Pal scores are oppositely correlated, to take the simple average we use the negative of the Pal score added to the Gi score. The highest score within each task is bolded. In the CINIC-10 columns, 'bn' stands batch-norm.

| | CIFAR-10 | | SVHN | CINIC-10 | | Oxford Flowers | Oxford Pets | Fashion MNIST | All Avg |
|---|---|---|---|---|---|---|---|---|---|
| | *VGG* | *NiN* | *NiN* | *Conv w/bn* | *Conv* | *NiN* | *NiN* | *VGG* | |
| AVG Gi intra ℓ0 & Gi intra ℓ1 | 0.06 | 26.25 | 18.26 | 16.16 | 12.09 | 15.77 | 7.80 | 5.10 | 12.69 |
| AVG Gi intra ℓ0 & Gi inter ℓ0 | 1.42 | 32.94 | 34.64 | 22.25 | 8.87 | 38.66 | 17.05 | 4.49 | 20.04 |
| AVG Gi intra ℓ0 & Gi inter ℓ1 | 4.29 | 27.89 | 20.38 | 18.11 | 9.64 | 17.69 | 8.73 | **15.88** | 15.33 |
| AVG Gi intra ℓ0 & Pal intra ℓ0 | 0.89 | 35.65 | **43.53** | 26.67 | 12.86 | **43.43** | 18.00 | 6.94 | **23.50** |
| AVG Gi intra ℓ0 & Pal intra ℓ1 | 0.24 | 15.71 | 12.59 | 10.03 | 13.09 | 7.15 | 5.22 | 8.35 | 9.05 |
| AVG Gi intra ℓ0 & Pal inter ℓ0 | 3.28 | **37.21** | 32.59 | 23.89 | 7.71 | 34.35 | **19.94** | 5.36 | 20.54 |
| AVG Gi intra ℓ0 & Mix intra ℓ0 | 0.13 | 16.33 | 20.56 | **30.17** | **18.15** | 29.83 | 7.87 | 13.22 | 17.03 |
| AVG Gi intra ℓ1 & Gi inter ℓ0 | 4.43 | 26.76 | 16.42 | 16.14 | 9.86 | 20.90 | 8.86 | 4.70 | 13.51 |
| AVG Gi intra ℓ1 & Mix intra ℓ0 | 0.40 | 9.15 | 22.71 | 23.89 | 9.28 | 18.83 | 1.83 | 10.92 | 12.13 |
| AVG Gi inter ℓ0 & Gi inter ℓ1 | 7.63 | 27.03 | 19.48 | 16.99 | 8.25 | 20.86 | 8.84 | 7.37 | 14.56 |
| AVG Gi inter ℓ0 & Mix intra ℓ0 | 2.99 | 7.67 | 19.00 | 24.59 | 18.04 | 15.05 | 5.01 | 0.98 | 11.66 |
| AVG Gi inter ℓ1 & Mix intra ℓ0 | 0.91 | 3.66 | 26.51 | 19.92 | 10.61 | 7.48 | 0.78 | 6.53 | 9.55 |
| AVG Pal intra ℓ0 & Pal inter ℓ0 | **24.84** | 29.70 | 14.04 | 1.64 | 3.45 | 14.84 | 2.13 | 4.89 | 11.94 |
| AVG Pal intra ℓ0 & Pal inter ℓ1 | 17.74 | 7.09 | 7.04 | 0.69 | 1.08 | 0.47 | 2.03 | 14.88 | 6.38 |
| *DBI*Mixup*[1] | *0.00* | *25.86* | *32.05* | *31.79* | *15.92* | *43.99* | *12.59* | *9.24* | *21.43* |

groups and correlates most strongly with generalization gap, especially when augmented training is not applied so models are less likely to be invariant to the test perturbation (as opposed to when

Table 7: Comparison of Conditional Mutual Information scores for various complexity measures across tasks. We present combinations of our measures using the simple product of two scores, and include the PGDL competition winner scores at the bottom. Note in this case, since Gi and Pal scores are oppositely correlated, we use the negative of the Pal score added to the Gi score. The highest score within each task is bolded. In the CINIC-10 columns, 'bn' stands batch-norm.

| | CIFAR-10 | | SVHN | CINIC-10 | | Oxford Flowers | Oxford Pets | Fashion MNIST | All Avg |
|---|---|---|---|---|---|---|---|---|---|
| | VGG | NiN | NiN | Conv w/bn | Conv | NiN | NiN | VGG | |
| PROD Gi intra $\ell 0$ & Gi intra $\ell 1$ | 0.05 | 26.76 | 17.92 | 16.36 | **12.05** | 14.90 | 7.70 | 4.92 | 12.58 |
| PROD Gi intra $\ell 0$ & Gi inter $\ell 0$ | 1.56 | 32.19 | 38.40 | 22.42 | 9.33 | **41.10** | 16.20 | 4.67 | 20.73 |
| PROD Gi intra $\ell 0$ & Gi inter $\ell 1$ | 1.77 | 29.05 | 27.02 | 19.08 | 9.76 | 26.62 | 10.33 | 10.72 | 16.79 |
| PROD Gi intra $\ell 0$ & Pal intra $\ell 0$ | 0.56 | 21.02 | 30.36 | 20.09 | 9.35 | 36.95 | 12.28 | 3.56 | 16.77 |
| PROD Gi intra $\ell 0$ & Pal intra $\ell 1$ | 0.80 | 21.78 | 30.58 | 20.26 | 9.21 | 36.19 | 10.82 | 3.54 | 16.65 |
| PROD Gi intra $\ell 0$ & Pal inter $\ell 0$ | 0.34 | 20.86 | 30.32 | 20.34 | 10.87 | 34.82 | 12.05 | 3.75 | 16.67 |
| PROD Gi intra $\ell 0$ & Mix intra $\ell 0$ | 0.93 | 29.90 | 40.23 | 17.46 | 6.67 | 40.51 | 12.76 | 4.84 | 19.16 |
| PROD Gi intra $\ell 1$ & Gi inter $\ell 0$ | 0.53 | 25.40 | 12.95 | 14.80 | 9.88 | 10.21 | 6.24 | 4.76 | 10.60 |
| PROD Gi intra $\ell 1$ & Mix intra $\ell 0$ | 0.26 | 15.38 | 10.15 | 4.28 | 8.39 | 2.81 | 1.93 | 5.48 | 6.09 |
| PROD Gi inter $\ell 0$ & Gi inter $\ell 1$ | 7.68 | 26.80 | 18.60 | 16.93 | 8.15 | 19.07 | 8.41 | 10.58 | 14.53 |
| PROD Gi inter $\ell 0$ & Mix intra $\ell 0$ | 4.69 | 30.90 | 13.94 | 10.78 | 0.74 | 19.91 | 6.44 | 4.19 | 11.45 |
| PROD Gi inter $\ell 1$ & Mix intra $\ell 0$ | **10.28** | 16.77 | 5.06 | 4.06 | 3.52 | 2.11 | 1.62 | 15.93 | 7.42 |
| PROD Pal intra $\ell 0$ & Pal inter $\ell 0$ | 1.71 | **35.77** | **41.58** | **25.14** | 9.50 | 38.92 | **18.41** | 5.61 | **22.08** |
| PROD Pal intra $\ell 0$ & Pal inter $\ell 1$ | 4.64 | 30.47 | 19.47 | 21.02 | 10.65 | 17.39 | 9.28 | **16.64** | 16.20 |
| *DBI\*Mixup*[1] | *0.00* | *25.86* | *32.05* | *31.79* | *15.92* | *43.99* | *12.59* | *9.24* | *21.43* |

Table 8: Comparison of Conditional Mutual Information scores for various complexity measures across tasks. We present combinations of our measures using the product of two scores plus the average of the same two scores, and include the PGDL competition winner scores at the bottom. Note in this case, since Gi and Pal scores are oppositely correlated, we use the negative of the Pal score added to the Gi score. The highest score within each task is bolded. In the CINIC-10 columns, 'bn' stands batch-norm.

| | CIFAR-10 | | SVHN | CINIC-10 | | Oxford Flowers | Oxford Pets | Fashion MNIST | All Avg |
|---|---|---|---|---|---|---|---|---|---|
| | VGG | NiN | NiN | Conv w/bn | Conv | NiN | NiN | VGG | |
| PROD+AVG Gi intra $\ell 0$ & Gi intra $\ell 1$ | 0.06 | 26.26 | 18.24 | 16.16 | 12.10 | 15.73 | 7.78 | 5.10 | 12.68 |
| PROD+AVG Gi intra $\ell 0$ & Gi inter $\ell 0$ | 1.40 | 32.85 | 34.91 | 22.29 | 8.94 | 38.94 | 17.03 | 4.46 | 20.10 |
| PROD+AVG Gi intra $\ell 0$ & Gi inter $\ell 1$ | 4.17 | 27.90 | 20.69 | 18.28 | 9.63 | 18.42 | 8.89 | 15.70 | 15.46 |
| PROD+AVG Gi intra $\ell 0$ & Pal intra $\ell 0$ | 0.79 | 17.24 | 32.09 | **33.10** | **16.09** | 40.89 | 22.60 | 11.47 | 21.78 |
| PROD+AVG Gi intra $\ell 0$ & Pal intra $\ell 1$ | 2.44 | 6.87 | 3.59 | 1.41 | 12.05 | 0.12 | 3.10 | 1.34 | 3.86 |
| PROD+AVG Gi intra $\ell 0$ & Pal inter $\ell 0$ | **11.85** | **36.14** | 15.51 | 24.13 | 4.06 | 27.34 | **26.74** | 5.42 | 18.90 |
| PROD+AVG Gi intra $\ell 0$ & Mix intra $\ell 0$ | 0.77 | 4.65 | 12.16 | 14.22 | 10.46 | 23.94 | 5.03 | 13.96 | 10.65 |
| PROD+AVG Gi intra $\ell 1$ & Gi inter $\ell 0$ | 4.31 | 26.68 | 16.15 | 16.05 | 9.79 | 19.56 | 8.65 | 4.72 | 13.24 |
| PROD+AVG Gi intra $\ell 1$ & Mix intra $\ell 0$ | 0.64 | 1.54 | 13.75 | 8.81 | 1.40 | 11.97 | 0.89 | 11.33 | 6.29 |
| PROD+AVG Gi inter $\ell 0$ & Gi inter $\ell 1$ | 7.63 | 27.01 | 19.44 | 16.98 | 8.24 | 20.66 | 8.74 | 7.39 | 14.51 |
| PROD+AVG Gi inter $\ell 0$ & Mix intra $\ell 0$ | 8.53 | 1.82 | 12.87 | 6.86 | 11.41 | 3.96 | 2.26 | 2.29 | 6.25 |
| PROD+AVG Gi inter $\ell 1$ & Mix intra $\ell 0$ | 1.97 | 2.65 | 18.78 | 5.12 | 2.37 | 3.59 | 1.97 | 11.29 | 5.97 |
| PROD+AVG Pal intra $\ell 0$ & Pal inter $\ell 0$ | 1.75 | 35.79 | **41.46** | 25.12 | 9.45 | 38.82 | 18.42 | 5.59 | **22.05** |
| PROD+AVG Pal intra $\ell 0$ & Pal inter $\ell 1$ | 4.73 | 30.42 | 19.34 | 20.91 | 10.65 | 17.15 | 9.13 | **16.62** | 16.12 |
| *DBI\*Mixup*[1] | *0.00* | *25.86* | *32.05* | *31.79* | *15.92* | *43.99* | *12.59* | *9.24* | *21.43* |

they are all trained to be invariant, which can potentially be achieved across hyperparameters if the perturbations are used in training).

Table 9: Comparison of Conditional Mutual Information scores for various complexity measures across tasks. We present combinations of our measures using the average of two scores converted to ranks across the task (i.e., rank all the scores for the task for the particular score, and add 1 go get the transformed score), and include the PGDL competition winner scores at the bottom. Note in this case, since Gi and Pal scores are oppositely correlated, we use the negative of the Pal score added to the Gi score. The highest score within each task is bolded. In the CINIC-10 columns, 'bn' stands batch-norm.

| | CIFAR-10 | | SVHN | CINIC-10 | | Oxford Flowers | Oxford Pets | Fashion MNIST | *All Avg* |
|---|---|---|---|---|---|---|---|---|---|
| | *VGG* | *NiN* | *NiN* | *Conv w/bn* | *Conv* | *NiN* | *NiN* | *VGG* | |
| AVG RANK Gi intra $\ell0$ & Gi intra $\ell1$ | 0.03 | 25.80 | 17.61 | 16.54 | 12.71 | 15.45 | 10.31 | 5.16 | 12.95 |
| AVG RANK Gi intra $\ell0$ & Gi inter $\ell0$ | 0.98 | 32.60 | 34.26 | 22.09 | 9.14 | 39.86 | 17.36 | 4.39 | 20.09 |
| AVG RANK Gi intra $\ell0$ & Gi inter $\ell1$ | 1.67 | 28.73 | 18.77 | 18.60 | 10.31 | 22.13 | 11.63 | **11.45** | 15.41 |
| AVG RANK Gi intra $\ell0$ & Pal intra $\ell0$ | 0.85 | 32.73 | **43.41** | **24.86** | 12.19 | **43.38** | 17.12 | 6.21 | **22.60** |
| AVG RANK Gi intra $\ell0$ & Pal intra $\ell1$ | 0.05 | 27.39 | 19.73 | 17.12 | **13.07** | 15.89 | 11.39 | 6.54 | 13.90 |
| AVG RANK Gi intra $\ell0$ & Pal inter $\ell0$ | 1.02 | **33.73** | 38.36 | 23.52 | 9.25 | 40.74 | **18.16** | 4.70 | 21.19 |
| AVG RANK Gi intra $\ell0$ & Mix intra $\ell0$ | 4.15 | 0.74 | 1.46 | 0.18 | 1.45 | 0.34 | 1.03 | 2.49 | 1.48 |
| AVG RANK Gi intra $\ell1$ & Gi inter $\ell0$ | 1.38 | 26.51 | 15.25 | 15.69 | 10.07 | 14.98 | 11.03 | 4.54 | 12.43 |
| AVG RANK Gi intra $\ell1$ & Mix intra $\ell0$ | 0.89 | 1.36 | 1.90 | 0.99 | 0.63 | 2.96 | 3.04 | 1.69 | 1.68 |
| AVG RANK Gi inter $\ell0$ & Gi inter $\ell1$ | 6.36 | 28.48 | 17.11 | 17.22 | 8.44 | 18.98 | 12.71 | 8.11 | 14.68 |
| AVG RANK Gi inter $\ell0$ & Mix intra $\ell0$ | 7.45 | 0.82 | 1.38 | 0.22 | 2.61 | 0.44 | 1.74 | 3.51 | 2.27 |
| AVG RANK Gi inter $\ell1$ & Mix intra $\ell0$ | 4.68 | 1.76 | 3.21 | 0.45 | 2.04 | 1.33 | 3.02 | 0.43 | 2.12 |
| AVG RANK Pal intra $\ell0$ & Pal inter $\ell0$ | **8.17** | 0.70 | 1.42 | 0.32 | 3.08 | 0.69 | 1.34 | 2.73 | 2.31 |
| AVG RANK Pal intra $\ell0$ & Pal inter $\ell1$ | 7.90 | 2.53 | 2.04 | 0.44 | 0.80 | 1.26 | 1.93 | 1.49 | 2.30 |
| *DBI\*Mixup*[1] | *0.00* | *25.86* | *32.05* | *31.79* | *15.92* | *43.99* | *12.59* | *9.24* | *21.43* |

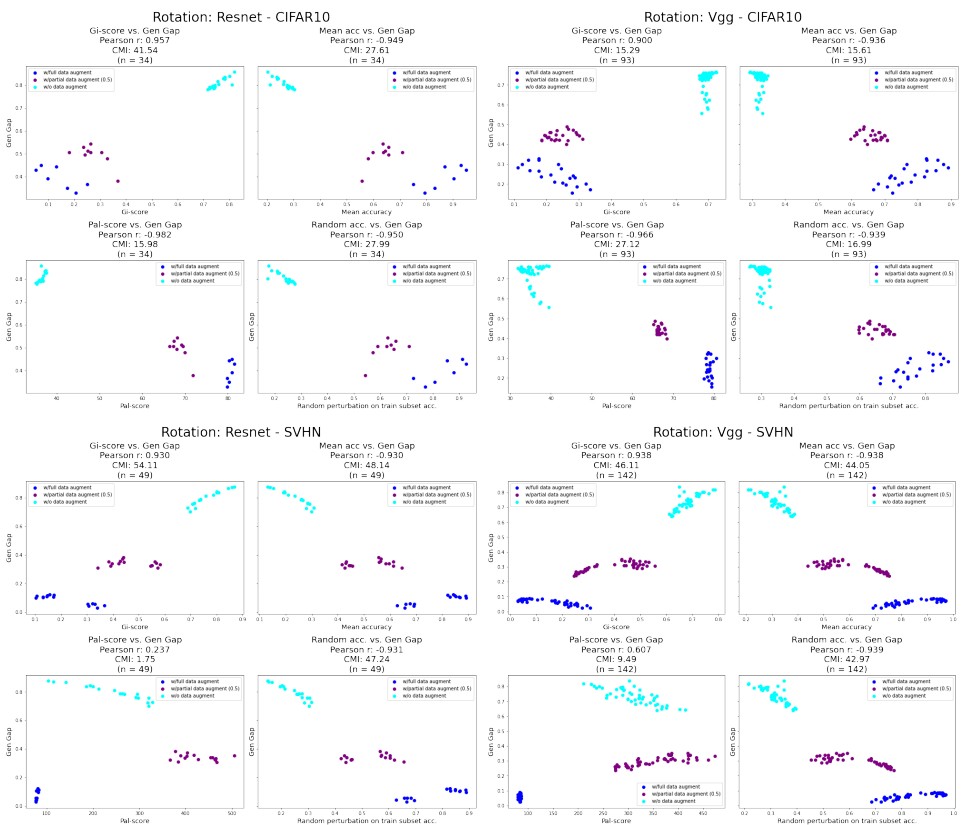

Figure 6: **Rotation:** Comparison of complexity measure and generalization gap for Resnet and VGG models trained on CIFAR-10 and SVHN to test how these measures predict generalization gap in the face of a rotation perturbation.

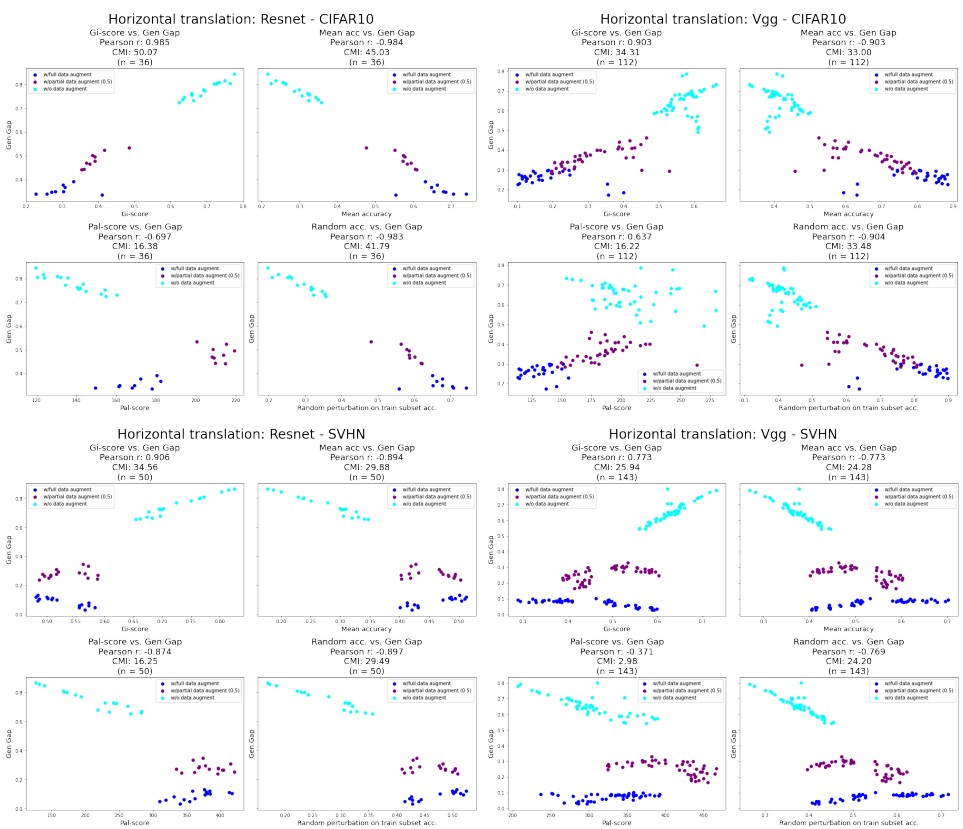

Figure 7: **Horizontal Translation:** Comparison of complexity measure and generalization gap for Resnet and VGG models trained on CIFAR-10 and SVHN to test how these measures predict generalization gap in the face of a horizontal translation perturbation.

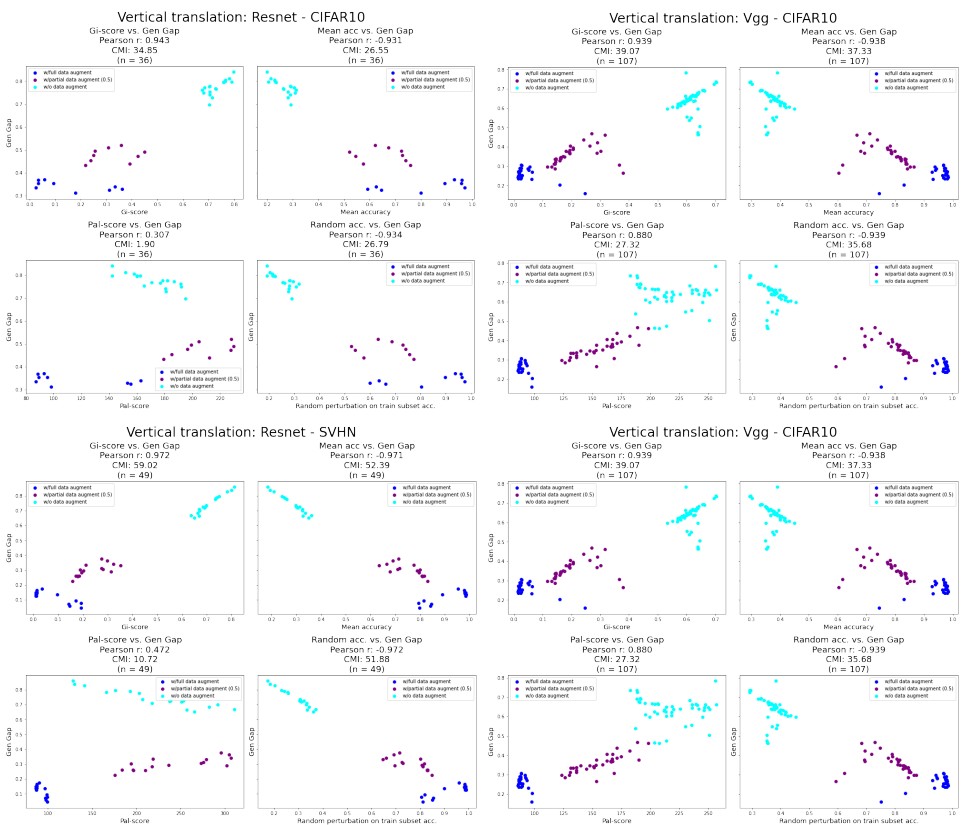

Figure 8: **Vertical Translation:** Comparison of complexity measure and generalization gap for Resnet and VGG models trained on CIFAR-10 and SVHN to test how these measures predict generalization gap in the face of a vertical translation perturbation.

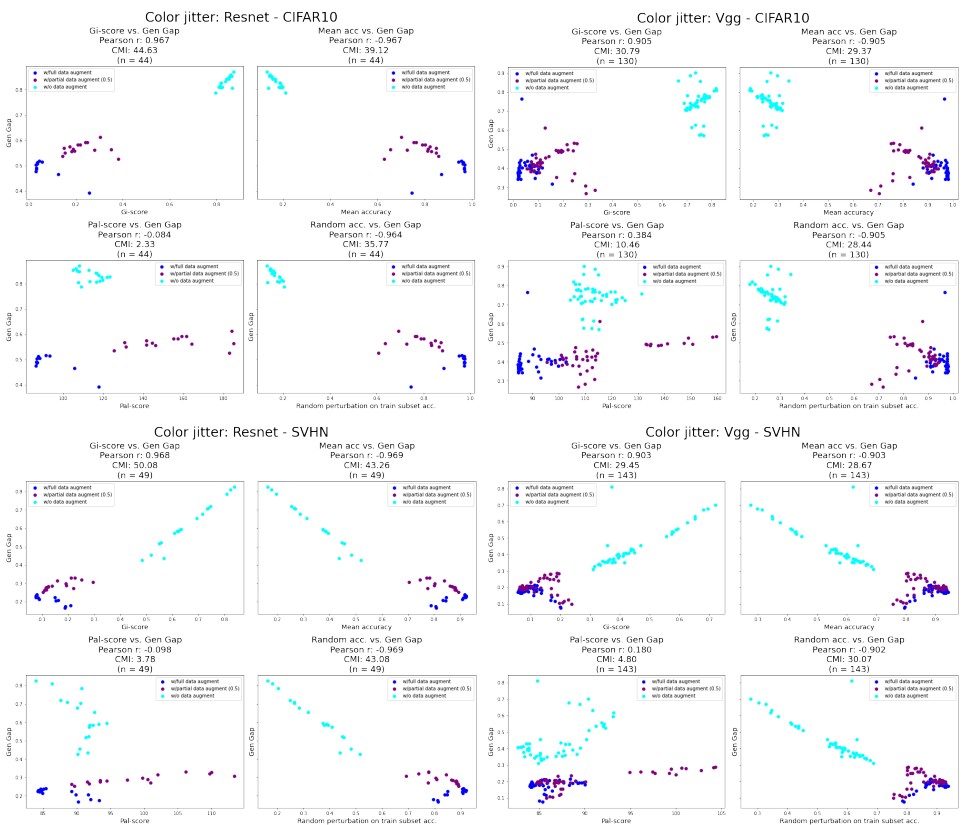

Figure 9: **Color-jittering:** Comparison of complexity measure and generalization gap for Resnet and VGG models trained on CIFAR-10 and SVHN to test how these measures predict generalization gap in the face of a color-jittering perturbation.