# OpenReview forum: "Predicting Deep Neural Network Generalization with Perturbation Response Curves"
_NeurIPS.cc/2021/Conference — NeurIPS 2021 Poster_

### Official Review · Reviewer_fdZt · 2021-07-09

**Rating:** 3
**Confidence:** 5

**Summary:**

This paper firstly uses Gini coefficient and Palma ratio together with PR curve for measuring generalization. The paper claims that their framework outperforms the state-of-the-art measures on most of the tasks in the PGDL competition and can capture invariance to input transformation for a trained network. Finally, the paper shows some experimental results on image classification.

**Limitations And Societal Impact:**

The usage of Gini and Palma statistics in the paper lacks analysis and explanation (e.g., which one to use and how the parameter should be set). Experimental results show that each type of their measure could perform much better even though some of them could be extremely bad. This phenomenon is not explained in this paper.

**Main Review:**

This work is about predicting deep neural network generalization but the authors focused too much on invariance. Maybe the authors should explain more about the relationship between generalization and invariance. They firstly use Gini and Palma statistics to measure generalization, which is novel, however, their usage is humble.

**Time Spent Reviewing:**

6 hours

---

> ### Author Response · Authors · 2021-08-10
> **Response to reviewer fdZt**
>
> >They firstly use Gini and Palma statistics to measure generalization, which is novel, however, their usage is humble.
>
>
> - We appreciate the reviewer's recognition of the novelty of our approach. In terms of usage, we believe that applying it to the corpus of trained networks from the NeurIPS PGDL competition is a valid use case given this competition was specifically designed to find measures that capture a network's ability to generalize from train to test data and was well received by the community.
>
>
> ------
>
>
> >Maybe the authors should explain more about the relationship between generalization and invariance... This work is about predicting deep neural network generalization but the authors focused too much on invariance.
>
>
> - We agree that there is a subtle nuance between broad generalization capability and invariance to a specific perturbation, which is why in our work we present these two uses cases separately and provide different experimentation to justify each individually.
>
>
> - However, invariance and generic generalization are in a sense very related. That is, if domain expertise indicates that a certain task should be invariant to a specific transformation (e.g., classifying images of cats under rotation), then higher degree invariance to that transformation amounts to generalizing to unseen data where that transformation is applied (following the classification example, full network invariance to rotation of images means that the network will generalize perfectly to any rotated images).
>
>
> - In revised work, we will make the distinction between the different use cases more explicit and also better explain how the two concepts are related.

---

> > ### Comment · Reviewer_fdZt · 2021-08-31
> > **Response to Paper5942 Authors**
> >
> > Dear authors，
> >
> > Thank you for replying to my questions.
> >
> > I do not deny that the originality of the paper is using Gini and Palma statistics to predict generalization, but I think this empirical guess without theoretical guarantee is relatively cheap. Even leaving aside the theoretical contribution of the paper, as far as the results found in the paper, this paper does not have enough evidence to demonstrate that this discovery can inspire researchers to further improve the algorithm. I think the exploration of this paper at the algorithmic and theoretical levels is still preliminary, more like an experimental report on the relationship between Palma statistics and NN's generalization performance. Therefore, I do not recommend that this paper be accepted by NeurIPS.

---

> ### Author Response · Authors · 2021-08-30
> **Looking forward to follow-up discussion**
>
> Dear Reviewer,
>
> Thank you again for your time and effort to produce your original review/comment. We've put considerable time and effort on our end to address your and other reviewers' concerns through new experiments and results along with individualized responses. It'd be appreciated if you can take a look at our responses to other reviewer concerns and our specific response to your individual review. We are eagerly awaiting your follow-up.

---

### Official Review · Reviewer_r6eX · 2021-07-15

**Rating:** 5
**Confidence:** 4

**Summary:**

The paper proposed two scores calculated from accuracy changes at varying levels of perturbation in training samples for predicting neural network generalization. The scores, Gi-score and Pal-score are inspired by the Gini and Palma index from economics.


**Limitations And Societal Impact:**

The authors described the limitations.
Societal impact: N/A.

**Main Review:**

The authors proposed a new framework, starting with evaluating trained networks on a subset of training data with varying levels of perturbation. They then construct the perturbation response (PR) curve and compute the Gi-score and Pal-score based on the PR curve. Two sets of experiments are done to demonstrate the usefulness of the scores (mainly Gi-score) in predicting neural network generalization and measuring network invariance.

The generalization prediction experiments are based on the PGDL competition, and in some cases, the performance of the proposed method surpasses the winning submission. Combinations of perturbation method (intra vs inter) and perturbed layer (input vs layer1) were used. The best performance was achieved with different combinations for each dataset. This seems to make the framework difficult to implement -- do people need to run all combinations (or even more) to reach a conclusion? Moreover, Gi-score almost always outperforms Pal-score. Is Pal-score more useful in any scenario? Although the authors draw connections with Gini coefficient and Palma ratio, they did not provide more intuition as to why these measures may be useful, or in what circumstances one measure is preferred to the other.  More discussions on this will help readers better appreciate the proposed framework.

In the generalization prediction experiments Gi-score is better than the value halfway on the PR curve (Mixup accuracy is $A_{0.5}^{(0)}$). For measuring invariance, Gini score is shown to be better than the mean accuracy and ($A_1^{(0)}$) at max perturbation. Again, more intuition and discussion on why Gi-score is a better choice will be helpful.

For presentation, while I appreciate it that the authors included pseudocode for Gini-score calculation, I’m afraid it does not provide more information than the shaded area in Fig.1. In fact I was surprised that Alg.2 was needed and had to read it in case I misunderstood the method. The space could be dedicated to more discussion and intuition.  I have a similar feeling about Alg.1.

The authors mentioned the lack of theoretical results a limitation of the paper. Indeed, the paper needs more intuition and discussion for readers to better appreciate the proposed framework.


**Time Spent Reviewing:**

3 hours

---

> ### Author Response · Authors · 2021-08-10
> **Response to reviewer r6eX (2/2):**
>
> >Although the authors draw connections with Gini coefficient and Palma ratio, they did not provide more intuition as to why these measures may be useful, or in what circumstances one measure is preferred to the other. More discussions on this will help readers better appreciate the proposed framework.
>
>
> - This is a fair point. In our revised work, we will add more detailed explanations as to why these economics-inspired statistics are useful and will provide more intuition for why we expected them to be beneficial.
>
>
> - Namely, we plan to highlight that the Gini coefficient is specifically designed to compare how the wealth distribution of a given economy compares to that of an idealized economy, where each percentage of the population has an equal share of the nation's income. In that way, the Gini coefficient does not characterize a nation's inequality by simply looking at the percentage of income that an individual percentile of the population holds: e.g. the top 10% or the bottom 50%iles. Rather, the Gini coefficient extracts aggregate information from the entire Lorenz curve which plots the wealth distribution across all percentiles of the population. In our work, this was the main inspiration for creating the Gi-score. We can measure a neural networks response to a certain magnitude of a transformation, but a priori, we do not know on which magnitude to focus. We hypothesize that looking at a single value of perturbation magnitude (which is essentially what the mixup type of approaches do) may not be ideal and/or informative, and that this can be viewed as a specific case of a more general framework we propose here, of evaluating a whole curve of perturbation magnitudes in different ways. We therefore propose to examine the spectrum of perturbation-responses and extract a more holistic view of how resilient a network is to a given transformation.
>
>
> - A key reasoning behind this is that even if two networks have the same accuracy after the maximum amount of perturbation (e.g., 0.5 for the case of Mixup) - that could be a final base level of accuracy deterioration (that multiple networks may share even), but the amount of deterioration before that point could still be different for different networks, and this may also be reflective of its generalization behavior.  For example, if a network's accuracy immediately fell off to the accuracy seen at maximum 0.5 interpolation perturbation at much smaller perturbations (e.g., 0.1), it is reasonable to believe that  this model is more brittle and less likely to generalize compared to a model that reached the same accuracy at 0.5 perturbation but only showed decreased  accuracy starting at  perturbation magnitudes very close to 0.5.  We provide an illustrative example of this from our experiments in the next comment response, showing this can indeed be the case.  Furthermore, the point at which even close-to-ideal models' accuracies would start to deteriorate could also be hard to pinpoint and depend on the data, further necessitating considering multiple points.
>
>
> - Now, given this justification, a plausible criticism of our work is: why compare to an idealized network? Why not just average the sampled points along the PR-curve? For this reason, in the measuring invariance experiments, we use this average PR as a baseline and find that the Gi-score produces more informative scores (higher CMI).  Additionally, a key point of our work is to leverage the full PR curve as opposed to an individual point, so averaging across the PR curve is also a new method that comes out of our framework - though it was found to be less effective in terms of the CMI scores in the cases we tested it.
>
>
> - In terms of when the Pal score is more informative than the Gi score, we believe that this will be domain / transformation specific and will require domain expertise. For general guidance as to what a practitioner should consider when deciding between the Gi and Pal scores, we again refer back to these scores' origins in economics. Specifically, the Palma ratio was introduced as an alternative to the Gini coefficient based on an empirical observation made by the economist José Gabriel Palma: the middle deciles 40-90% often account for roughly 50% of a nations income across various economies. This 'middle-class's share' is therefore less informative in distinguishing the income inequality of two economies as this share seems to be consistent for different nations. The Pal-score in our framework would be deployed in similar situations, namely if a domain expert observed that the PR curves for a certain transformation have a region that does not vary much by dataset/architecture, then it would be sensible to ignore this region when calculating generalization scores from the PR curves.
>
>
> >In the generalization prediction experiments Gi-score is better than the value halfway on the PR curve (Mixup accuracy is $A_{(0.5)}^{(0)}$). For measuring invariance, Gini score is shown to be better than the mean accuracy and ($A_{(1)}^{(0)}$) at max perturbation. Again, more intuition and discussion on why Gi-score is a better choice will be helpful.
>
>
> - As mentioned above, a key intuition behind this is the idea that using a single point in the PR curve (e.g., as mixup does) may not fully capture the differences in PR curves - for example, two PR curves might end up with the same or close to the same value at a particular perturbation amount, but may differ greatly before that perturbation amount.  In the case of accuracy, one may deteriorate in accuracy much sooner and more quickly than the other, even if they end up with roughly the same accuracy for the maximum perturbation magnitude (or a particular perturbation magnitude) such as 0.5 for our interpolation perturbations.  We found this to indeed be the case in some instances, by looking at the PR curves in the experiment results.
>
>
> - We looked at the experiment results in detail to provide some clearly illustrative examples to help better explain this, and we will include them in the revision.
>
>      - E.g., we found examples of pairs of models that had significantly different generalization gap, but for which mixup scores were roughly the same, and Gi-score was significantly different and reflective of the difference in generalization gap.  Looking at the PR curves, we see the higher generalization gap model's PR curve deteriorating more quickly / being under the lower generalization gap model's PR curve, despite having roughly the same accuracy at $\alpha = 0.5$, validating our hypothesis.
>
>    - E.g., for one particular illustrative example for SVHN, we compared PR curves and scores for two models, model 1006 with normalized generalization gap 0.22, and model 243 with normalized generalization gap 0.36 (where we normalized generalization gap to be between 0 and 1 across all the models on the data).  We found in this case the normalized mixup score (again normalized across the mixup scores of models trained on this data set) to be 0.26 and 0.27 respectively vs. normalized gini scores to be 0.22 and 0.36 respectively.  In this case the gini scores aligned much more closely to the generalization gap, and there is a clear gap in the PR curves between the higher and lower generalization gap models, despite the final value of the curve at 0.5 being roughly equal.  As a result, Mixup is not able to capture this difference between the PR curves and model sensitivity to perturbation, but our proposed framework and Gi-score does.  This figure can be seen here: https://www.dropbox.com/s/cu72rcnb7qm23bk/pr_curve_gini_vs_mixup_illustration.jpg?dl=0
>
>
> ------
>
>
> >The authors mentioned the lack of theoretical results a limitation of the paper. Indeed, the paper needs more intuition and discussion for readers to better appreciate the proposed framework.
>
>
> - One of the major motivations for the PGDL competition is that theoretical bounds often do not hold up in practice. Therefore the organizers wanted to find empirical measures that perform well in predicting generalization gap. We believe our work fits this need. As discussed above, we will provide more intuition as to how we derived the use of our framework in the revised version, while theoretical underpinnings will be left for future work.  Further, we hope our approach and promising experimental results will help inspire future work and study along these directions.

---

> ### Author Response · Authors · 2021-08-10
> **Response to reviewer r6eX (1/2)**
>
> >The authors proposed a new framework.
>
>
> - We are happy to see that the reviewer finds the proposed framework novel.
>
> -----
>
> >Combinations of perturbation method (intra vs inter) and perturbed layer (input vs layer1) were used. The best performance was achieved with different combinations for each dataset. This seems to make the framework difficult to implement -- do people need to run all combinations (or even more) to reach a conclusion?
>
>
> - We appreciate the reviewer's feedback here. Table 1 was intended to provide a detailed breakdown of results across the PGDL competition framework.  To address the point raised by the reviewer, we have performed detailed experiments to derive  clearer guidance (as described in the tables below) on which score / combination of scores is most promising for each specific network architecture type in order to make our work more actionable for practitioners. Please see the summary tables below, which will be included in the revised version.
>
>
> - The high level summary from these new and detailed experiments is that Gi scores for both intra and inter-class mixups at the input layer appear to be most informative across architectures, except for VGG-type architectures where scores calculated from the first hidden layer are most informative. Future work will explore reasons why different architectures are better explained by the scores at either input or hidden layers.
>
>
> - Furthermore, we note that Gi intra $\ell=0$ alone averaged over all datasets is comparable to the average score of the PGDL competition winner (i.e., when averaged over 20 random runs with 180 batches of the data, Gi intra $\ell=0$ average score across all datasets is 21.41 vs. 21.43 for the competition winner), and we'll update the table with these results.
>
>
> - Additionally, systematic combination of individual scores also revealed several combinations of our scores yielding even better results overall, and we will include these overall results and combined score results in the revised version as well.
>
>
> - In particular, overall we get the best results by using PCA to combine Gi intra $\ell = 0$ with mixup  - i.e., overall score of 23.83 (std. dev. = 1) vs. 21.43 for the competition winner.  This could correspond to the setting in which we have a set of trained models for a dataset and want to evaluate their relative generalization performance.  Other combinations of scores also improve over the competition winner and can provide go-to rules.  For example, the product of Pal *inter* $\ell = 0$ and Pal *intra* $\ell = 0$ yields an overall average of 22.08 (std. dev. = 0.46), again having a higher average score across all datasets than the competition winner.  We'll include full results for the combinations in the revision.
>
>
> **NiN architectures:** Gi-scores for both inter- and intra- class mixup on inputs are most informative
>
>
> |                | CIFAR-10 | SVHN  | Oxford Flowers | Oxford Pets | Average |
> |----------------|----------|-------|----------------|-------------|---------|
> | Gi inter $\ell=0$ |    34.78 | 26.86 |          33.35 |       17.80 |   28.20 |
> | Gi intra $\ell=0$ |    31.73 | 40.99 |          40.56 |       16.80 |   32.52 |
> | PGDL winner|   25.86  | 32.05 |           43.99 |      12.59 |    28.62 |
>
>
>
> **VGG architectures:** Gi and Pal scores on inter-class mixup from layer 1 representations are most informative
>
>
> |                 | CIFAR-10 | Fashion MNIST | Average |
> |-----------------|----------|---------------|---------|
> | Gi inter $\ell=1$  |     7.69 |         16.12 |   11.91 |
> | Pal inter $\ell=1$ |     7.10 |         14.43 |   10.77 |
> | PGDL winner |           0 |           9.24 |     4.62 |
>
>
>
>
> **Fully convolutional architectures:** Both Gi and Pal scores for intra- class mixup on inputs are most informative.
> However, there is still a gap compared to the PGDL competition winner approach on these network types, and mixup alone works well on this dataset - it seems for this particular set of architectures or dataset, the deterioration of accuracy at larger perturbations / closer to the max may be more important to focus on / weight higher in determining generalization. However, even in this case we find that information from other parts of the perturbation response curve can still be useful.  We found that by combining our Gi intra ell=0 score (that summarizes the whole PR curve) with mixup (that focuses on the end of the PR curve), via PCA, we get the best results across these data sets / architectures.  Future work in loves investigating and understanding why generalization for this particular architecture / dataset depends more heavily on the end of the PR curve as opposed it earlier parts of the curve.
>
>
>
>
> |                 | CINIC-10 (w/bn) | CINIC-10 (w/o bn) | Average |
> |-----------------|-----------------|-------------------|---------|
> | Gi intra $\ell=0$  |           22.80 |             11.49 |   17.15 |
> | Pal intra $\ell=0$ |           24.38 |             10.93 |   17.66 |
> | PGDL winner|            31.79 |             15.92 |  23.855|
> |-----------------|-----------------|-------------------|---------|
> | Mixup             |             30.3 |             19.51 |  24.905|
> |-----------------|-----------------|-------------------|---------|
> | PCA combine Gi intra $\ell = 0$ + Mixup|           33.76 |             20.33 |    27.04|
>
> ------
>
>
> >Moreover, Gi-score almost always outperforms Pal-score. Is Pal-score more useful in any scenario?
>
>
>
>
> - This is a good observation. Since Palma ratio has been shown to be a useful extension / variation of the Gini score in measuring economic inequality we wanted to explore it here as well.
>
>
>
>
> - As seen in the our breakdown above by architecture, the Pal score is often comparable to the Gi score and in some cases more informative.
>
>
>
>
> - Finally, once someone has used our framework to calculate the PR curves, both the Gi and Pal scores are easily extracted. Therefore, if there is a setting where a data scientist or machine learning practitioner knows that certain regions of the PR curve are more informative than others, then the Pal score will be more useful.
>
>
>
>
> - We will add this discussion in the revised version.

---

> ### Author Response · Authors · 2021-08-30
> **Looking forward to follow-up discussion**
>
> Dear Reviewer,
>
> Thank you again for your time and effort to produce your original review/comment. We've put considerable time and effort on our end to address your and other reviewers' concerns through new experiments and results along with individualized responses. It'd be appreciated if you can take a look at our responses to other reviewer concerns and our specific response to your individual review. We are eagerly awaiting your follow-up.

---

> ### Comment · Reviewer_r6eX · 2021-08-31
> **Thanks for the detailed response**
>
> Dear authors,
>
> Thank you for the detailed responses to all my questions and the additional information you provided. I have read it carefully. However, I still think the paper in the current state is preliminary and not ready for publication. I would recommend you to organize the results with more focus on the configuration or method that you would like to recommend, and explain in depths why it is useful, either theoretically or empirically. Your current explanation on why these metrics used in Economics may be a good fit for examining neural networks feels still superficial -- the entire curve may be more useful than a single point.
>
> I agree that your idea may be useful however in the meantime feel the paper needs a substantial amount of work to be considered for publication.
>
> Thanks!

---

### Official Review · Reviewer_wkSZ · 2021-07-16

**Rating:** 8
**Confidence:** 3

**Summary:**

The authors propose a new framework to evaluate the generalizability of trained networks. The framework is based on evaluating the perturbation response curve on intra and inter-class sample mixup. It uses two novel statistical measures: the Gi-score and the Pal-score to predict the generalization gap. The framework is able to achieve a state-of-the-art predictive measure for several tasks in the PGDL competition and can be used to capture invariants to parametric input transformations.

**Ethical Concerns:**

To my knowledge, there are no ethical issues with the paper

**Limitations And Societal Impact:**

The authors have adequately addressed the limitations of their work.

**Main Review:**

Originality:
While closely resembling [1, 2], the proposed method includes some substantially novel and original contributions including inter-class sample mixup as well as a spectrum of parameterized perturbations.

Quality:
The paper successfully supports the predictive generalization claim through experiments based on the PGDL competition dataset and the predictive invariance claim based on a custom dataset built around the CIFAR-10 and SVHN datasets. It would have been additionally beneficial for the authors to add experiments with more challenging datasets like [3, 4] which were built to evaluate the generalizability and robustness of the networks to domain shift.

Clarity:
The paper is well written and clear.

Significance:
The paper shows significant improvement to the predictive generalizability problem which is very important for the goal of making better, more generalizable models. Also, the framework is successfully able to learn task-dependent invariances, which is a critical task for many machine learning applications. One potential limitation of this framework is that it has not been validated on non-image modalities.

References:
- [1] Natekar, P., & Sharma, M. (2020). Representation based complexity measures for predicting generalization in deep learning. arXiv preprint arXiv:2012.02775.
- [2] Kashyap, D., & Subramanyam, N. (2021). Robustness to Augmentations as a Generalization metric. arXiv preprint arXiv:2101.06459.
- [3] Barbu, A., Mayo, D., Alverio, J., Luo, W., Wang, C., Gutfreund, D., ... & Katz, B. (2019).  Objectnet: A large-scale bias-controlled dataset for pushing the limits of object recognition models. In Advances in Neural Information Processing Systems 32, pages 9448–9458. 2019.
- [4] Koh, P. W., Sagawa, S., Xie, S. M., Zhang, M., Balsubramani, A., Hu, W., ... & Liang, P. (2021, July). Wilds: A benchmark of in-the-wild distribution shifts. In International Conference on Machine Learning (pp. 5637-5664). PMLR.

**Time Spent Reviewing:**

3

---

> ### Author Response · Authors · 2021-08-10
> **Response to reviewer wkSZ**
>
> > the proposed method includes some substantially novel and original contributions
>
>
> - We thank the reviewer for acknowledging the novelty and originality of our work.
>
>
> -------
>
> >It would have been additionally beneficial for the authors to add experiments with more challenging datasets like [3, 4] which were built to evaluate the generalizability and robustness of the networks to domain shift.
>
> - We focused on the datasets from the PGDL challenge and predicting generalization, because as the organizers note (https://arxiv.org/abs/2012.07976)  "Generalization is one the most fundamental question of machine learning. A principled understanding of generalization can provide theoretical guarantees for machine learning algorithms, which makes deep learning more accountable and transparent, and is also desirable in safety critical applications."  The competition was well-received and had many teams participating, indicating that datasets made available in PGDL are valuable. Moreover, the PGDL competition provides a defined framework for evaluating the proposed generalization measures.
>
> - The reviewer raises a valid point, a great extension to our work is to see if it is applicable in domain shift settings. We believe that our 'measuring invariance' experiments are a first step at testing our framework in domain shift settings. In [3], some camera angle changes are considered 'domain shifts', inline with our experiments on rotation. In the future, we plan to further explore these more challenging datasets and other instances of domain shift.
>
> -----------
>
> >One potential limitation of this framework is that it has not been validated on non-image modalities.
>
> - This is a good point. For our initial exploration, we focused on image modality, as in the PGDL competition, which provides a very focused and useful code base with pre-trained models for estimating conditional mutual information. As far as we are aware, many related works on measuring network generalization have also focused solely on the image modality.
>
> - In line with the reviewer's comments, we plan to include results on an additional modality in the revised version of this work (considering audio, time series, or tabular). Testing these other modalities will require training  a new set of neural networks from scratch on the new datasets, which is time-consuming and could not be completed during the 5 days of rebuttal period.

---

> > ### Comment · Reviewer_wkSZ · 2021-08-31
> > **Response to Authors**
> >
> > Thank you for the response to my questions. I appreciate that many of the experiments I have suggested take a long time to run and would not be feasible to complete in the discussion period. I think the use of the Gi-score and Pal-score as a measure of generalization is a significant contribution. The scores could be extremely useful for evaluating and improving generalization performance for novel methods. I will maintain my score of 8.

---

> ### Author Response · Authors · 2021-08-30
> **Looking forward to follow-up discussion**
>
> Dear Reviewer,
>
> Thank you again for your time and effort to produce your original review/comment. We've put considerable time and effort on our end to address your and other reviewers' concerns through new experiments and results along with individualized responses. It'd be appreciated if you can take a look at our responses to other reviewer concerns and our specific response to your individual review. We are eagerly awaiting your follow-up.

---

### Decision · Program_Chairs · 2021-09-27

**Decision:**

Accept (Poster)

**Comment:**

The paper presents the idea of perturbation response curve and by combining that Gi-score and the Pal-score, proposes a new way for predicting the generalization gap. The method is able to achieve impressive score on several tasks in the PGDL competition. Reviewer wkSZ provides a strong support and finds the submission well-written and the result significant. Reviewer r6eX give a borderline rating. One of the concerns of this reviewer was related to the requirement of having different hyperparameters for the proposed scheme for each dataset, in order to achieve the best prediction result. In the rebuttal, the authors performed additional analysis and reported a common setting that performs well on most of the PGDL tasks. In addition, reviewer r6eX about the intuition behind Gini coefficient and Palma ratio and the reason they are useful, which is again explained by the authors in their response, and planned to be included in the final revision. Reviewer fdZt had stronger reservation against the paper, and was mainly seeking more theoretical justification for the proposed scheme. Overall, the reviews were diverge, but I myself lean to accept the submission, because I think the paper excels in the empirical side (on various PGDL competition tasks) and it is interesting enough of a result without need for a theory component, and the findings of this work can provide new insights about generalization, and be a stepping-stone for deeper analysis in the future. I recommend accept and ask authors to incorporate the clarifications/explanations that provided here in the paper as well.